

# Impurities in a one-dimensional Bose gas: the flow equation approach

Fabian Brauneis[1, 2], Hans-Werner Hammer[1, 3],
Mikhail Lemeshko[2] and Artem G. Volosniev[2,★]

**1** Technische Universität Darmstadt, Department of Physics,
Institut für Kernphysik, 64289 Darmstadt, Germany
**2** Institute of Science and Technology Austria, Am Campus 1, 3400 Klosterneuburg, Austria
**3** ExtreMe Matter Institute EMMI, GSI Helmholtzzentrum für Schwerionenforschung GmbH,
64291 Darmstadt, Germany

★ artem.volosniev@ist.ac.at

## Abstract

A few years ago, flow equations were introduced as a technique for calculating the ground-state energies of cold Bose gases with and without impurities [1,2]. In this paper, we extend this approach to compute observables other than the energy. As an example, we calculate the densities, and phase fluctuations of one-dimensional Bose gases with one and two impurities.

For a single mobile impurity, we use flow equations to validate the mean-field results obtained upon the Lee-Low-Pines transformation. We show that the mean-field approximation is accurate for all values of the boson-impurity interaction strength as long as the phase coherence length is much larger than the healing length of the condensate.

For two static impurities, we calculate impurity-impurity interactions induced by the Bose gas. We find that leading order perturbation theory fails when boson-impurity interactions are stronger than boson-boson interactions. The mean-field approximation reproduces the flow equation results for all values of the boson-impurity interaction strength as long as boson-boson interactions are weak.



# 1 Introduction

The flow equation approach is an *ab-initio* method for solving many-body problems [3]. A related method, the in-medium similarity renormalization group (IM-SRG), was recently developed and successfully used in nuclear physics (see, e.g., Refs. [4, 5])[1]. The main concept behind both methods is the same. Therefore, we use 'IM-SRG' and 'flow equations' interchangeably in this paper.

IM-SRG was recently extended to cold Bose gases [1, 2]. It was tested by calculating the ground-state energies of the Lieb-Liniger model and a one-dimensional (1D) Bose gas with an impurity atom ('Bose polaron') [1, 2]. Those works demonstrate that flow equations allow one to go beyond mean-field approximation without relying on many-body perturbation theory. In the present work, we use IM-SRG to calculate the density and phase fluctuations of the Bose gas. Motivated by recent cold-atom experiments [9, 10], we focus on the 1D Bose-polaron problem. This problem is of current theoretical interest, see Refs. [2, 11–25], which provide us with data for benchmarking and interpreting some of our IM-SRG results.

It has been noticed that the mean-field approximation (MFA) in a frame co-moving with the impurity can accurately describe the self-energy of the impurity in a weakly-interacting Bose gas [2, 17, 19, 22]. This observation is somewhat counter-intuitive, since strong phase fluctuations in one spatial dimension require a beyond-mean-field treatment. A counterargument to this point can be based on the observation that (when solving a Bose-polaron problem) one is

---

[1]Note that the word 'in-medium' in the name of the method is used to separate the IM-SRG from the standard SRG approaches, which are used in nuclear physics to 'soften' nuclear forces before using them in *ab initio* methods (see, e.g., Refs. [6–8]), such as a no-core shell model.

usually interested only in what happens to a Bose gas in the vicinity of the impurity, and therefore, the absence of long-range order is not necessarily relevant. Therefore, the mean-field approach can be useful (cf. [26]) as long as the phase coherence length, $\xi e^{\sqrt{\pi^2/\gamma}}$, is larger than the length scale associated with the polaron, which is of the order of $\xi$ [2]. Here $\xi$ is the healing length of the condensate, and $\gamma$ is the dimensionless Lieb-Liniger parameter which characterizes the boson-boson interaction strength, see Sec. 3. This argument implies that as long as $e^{\sqrt{\pi^2/\gamma}} \gg 1$ one can use the MFA to study the Bose-polaron problem. The discussion above is based on perturbation theory, and further work is needed for its rigorous proof. Here, we use the IM-SRG method to provide numerical evidence for its validity.

In this work, we benchmark mean-field results against those obtained with IM-SRG, and find that the MFA can describe the density of a Bose gas with an impurity particle accurately. To confirm the absence of boson-boson entanglement in the frame co-moving with the impurity, we calculate the phase fluctuations. They turn out to be negligible for the considered systems. Finally, we use the Born-Oppenheimer approximation to estimate the potential between impurities supported by a Bose gas. Our IM-SRG results show that the mean-field approximation is useful to study mesoscopic and large systems with two impurities. In particular, the MFA allows one to calculate the induced impurity-impurity interaction potential beyond first-order perturbation theory in a simple manner, i.e., without including any information about beyond-mean-field boson-boson correlations.

The paper is organized as follows: In Sec. 2, we give a short review of the IM-SRG, and explain how to calculate observables using this method. In Sec. 3, we discuss a Bose gas with a single impurity assuming repulsive boson-impurity interactions. There, we benchmark the IM-SRG results for the density against the exact Bethe-ansatz solution. Furthermore, we calculate the density and phase fluctuations of the Bose gas for systems that do not allow for an exact analytic treatment. In Sec. 4, we consider a Bose gas with two impurities, and calculate induced impurity-impurity interactions. We show that two impurities attract each other for repulsive impurity-boson interactions, whereas two impurities repel each other if one impurity attracts and another repels bosons, in accord with Refs. [23, 27]. We summarize our results and give an outlook in Sec. 5. Further details on the IM-SRG method in our implementation are given in Appendix A. For convenience of the reader, we present some additional results for a system with attractive boson-impurity interactions in Appendix B.

## 2 In-Medium Similarity Renormalization Group

For convenience of the reader, we shall present in this chapter the main ingredients of the IM-SRG method for bosons, see also Ref. [1], Appendix A, and Fig. 1.

### 2.1 Flow equations

The IM-SRG is an extension of the SRG [3, 28, 29] based upon the flow equation

$$\frac{dH}{ds} = [\eta, H], \tag{1}$$

which transforms the Hamiltonian matrix into a block-diagonal form, i.e., it decouples the "ground-state" matrix element from all excitations (see Fig. 1). The flow equation is defined once the initial condition, $H(s = 0)$, and the generator of the transformation, $\eta$, are specified. It is worth noting that Eq. (1) is equivalent to the unitary transformation $H(s) = U(s)H(s = 0)U^\dagger(s)$, assuming that the antihermitian operator $\eta$ and the unitary operator $U$ are connected as

$$\eta(s) = \frac{dU(s)}{ds}U^\dagger(s). \tag{2}$$

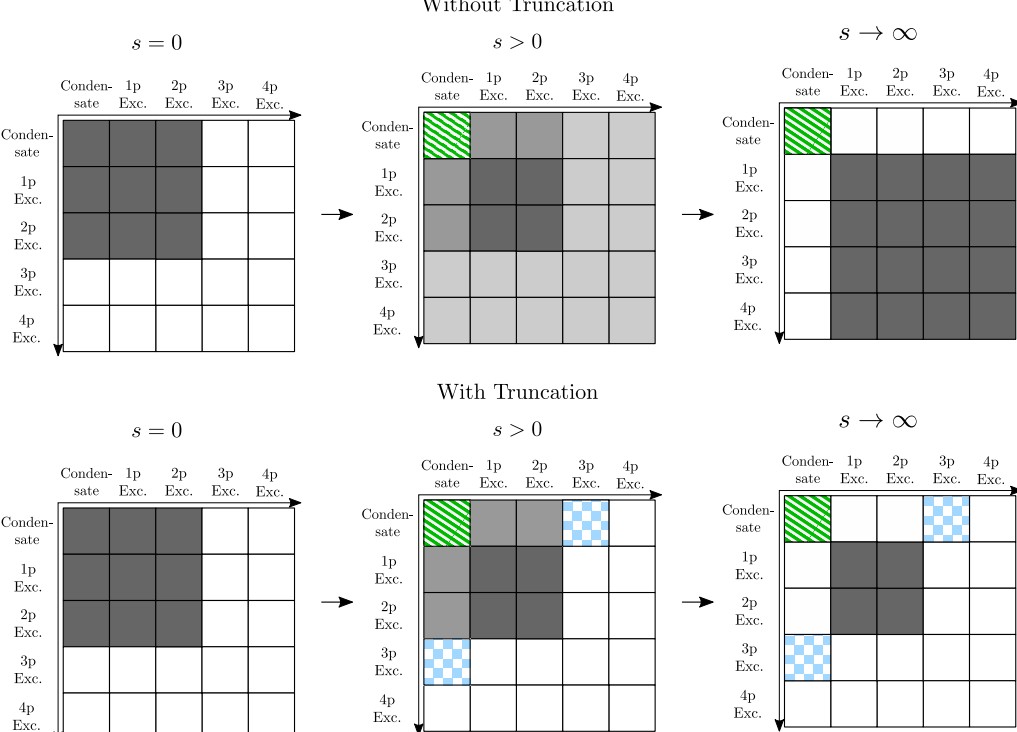

Figure 1: Illustration of the action of the flow equation (1) on the Hamiltonian. The Hamiltonian matrix is unitarily transformed to a block-diagonal form, such that the ground state (hatched green) becomes decoupled. The upper row illustrates the exact transformation without a truncation. Note that many-body excitations appear during the flow. The bottom row illustrates a truncation scheme adopted to circumvent this problem. The induced three-body terms (checkered blue) can be estimated and used to evaluate the accuracy of the truncation scheme. The excitations are defined with respect to the adopted normal ordering, see Sec. 2.2.

We prefer to write the unitary transformation in the form of Eq. (1) because it allows us to choose the operator $\eta(s)$ during the flow, i.e., for every parameter $s$, and, hence, to steer the flow in the desired direction. In our work, $\eta(s)$ is chosen from the matrix elements that describe the couplings between the 'condensate' and its excitations such that these couplings become weaker as the flow progresses, see Fig. 1. A detailed construction of $\eta(s)$ is presented in Appendix A.

In practice, the following steps constitute the IM-SRG method: (i) find a one-body basis to write a Hamiltonian matrix in second quantization (see Appendix A.2), (ii) find a reference state to normal order the operators (see the next subsection), (iii) solve the flow equation (1) (we use the explicit Runge-Kutta method 5(4)).

## 2.2 Normal ordering

In general, it is impossible to solve Eq. (1) for a many-particle system without approximations. The complexity is due to the commutator $[\eta, H]$: It leads to many-body terms, which are not present in the initial Hamiltonian $H(s = 0)$. To solve Eq. (1), the many-body terms must be truncated at some order. To define a truncation hierarchy, we write the Hamiltonian $H$ in second quantization using normal ordering with respect to a reference function, $\Psi_{\text{ref}}$, which is a product state, see the next subsection. Upon normal ordering, we truncate three-body excitations and beyond, see Fig. 1. To estimate the introduced truncation error, we use

the three-body elements and second order perturbation theory for matrices, see Fig. 1 and Appendix A.1.

To normal order a Hamiltonian with one- and two-body operators, we define the contractions[2], following Ref. [1]

$$: a^\dagger_{\alpha_1} a_{\alpha_2} : = a^\dagger_{\alpha_1} a_{\alpha_2} - I C_{\alpha_1 \alpha_2}, \tag{3}$$

$$\begin{aligned} : a^\dagger_{\alpha_1} a^\dagger_{\alpha_2} a_{\alpha_3} a_{\alpha_4} : = a^\dagger_{\alpha_1} a^\dagger_{\alpha_2} a_{\alpha_3} a_{\alpha_4} - I C_{\alpha_1 \alpha_2 \alpha_3 \alpha_4} \\ - \frac{N-1}{2N}(1 + P_{\alpha_1 \alpha_2})(1 + P_{\alpha_3 \alpha_4}) C_{\alpha_2 \alpha_3} : a^\dagger_{\alpha_1} a_{\alpha_4} :, \end{aligned} \tag{4}$$

where $a^\dagger_\alpha$ is a bosonic creation operator, $C_{\alpha_1 \alpha_2} = \langle \Psi_{\text{ref}} | a^\dagger_{\alpha_1} a_{\alpha_2} | \Psi_{\text{ref}} \rangle$ and $C_{\alpha_1 \alpha_2 \alpha_3 \alpha_4} = \langle \Psi_{\text{ref}} | a^\dagger_{\alpha_1} a^\dagger_{\alpha_2} a_{\alpha_3} a_{\alpha_4} | \Psi_{\text{ref}} \rangle$. The parameter $N$ is the number of bosons. $I$ is the identity operator; the operator $P_{\alpha_1 \alpha_2}$ swaps the indices $\alpha_1$ and $\alpha_2$.

A generic Hamiltonian with one- and two-body operators,

$$H = \sum_{i,j} A_{ij} a^\dagger_i a_j + \frac{1}{2} \sum_{i,j,k,l} B_{ijkl} a^\dagger_i a^\dagger_j a_k a_l , \tag{5}$$

reads in the normal ordered prescription as

$$H = \epsilon N I + \sum_{i,j} f_{ij} : a^\dagger_i a_j : + \frac{1}{2} \sum_{i,j,k,l} \Gamma_{ijkl} : a^\dagger_i a^\dagger_j a_k a_l :, \tag{6}$$

with

$$\epsilon N = \sum_{i,j} A_{ij} C_{ij} + \frac{1}{2} \sum_{i,j,k,l} B_{ijkl} C_{ijkl} , \tag{7}$$

$$f_{\alpha_1 \alpha_2} = A_{\alpha_1 \alpha_2} + \frac{N-1}{N} \sum_{i,j} B_{\alpha_1 i j \alpha_2} C_{ij} , \tag{8}$$

$$\Gamma_{\alpha_1 \alpha_2 \alpha_3 \alpha_4} = B_{\alpha_1 \alpha_2 \alpha_3 \alpha_4} . \tag{9}$$

$\epsilon N = E$ is the energy of the ground state, $f_{\alpha_1 \alpha_2}$ ($\Gamma_{\alpha_1 \alpha_2 \alpha_3 \alpha_4}$) describes one-particle (two-particle) excitations.

## 2.3 Reference state

The reference state, $\Psi_{\text{ref}}$, should approximate an eigenstate (here the ground state) of the Hamiltonian well, otherwise the IM-SRG transformation cannot map $\Psi_{\text{ref}}$ onto the exact state. Since we are interested in ground-state properties of a bosonic system, it is logical to use a product state as a reference state, i.e.,

$$\Psi_{\text{ref}}(x_1, ..., x_N) = \prod_{\alpha=1}^{N} f(x_\alpha), \tag{10}$$

where $x_\alpha$ is the coordinate of the $\alpha$th boson, and $f$ is some function whose form we discuss below. The choice of a product-state ansatz is natural for cold Bose gases with macroscopic population of a single mode, i.e., with a large condensate fraction. However, for an interacting one-dimensional Bose gas, the reference state of Eq. (10) is in general not accurate. In the thermodynamic limit, correlations in the 1D Bose gas decay algebraically, precluding condensation [30–34]. However, as our analysis below shows, the product state is a useful starting point for analyzing properties of Bose polarons.

---

[2]We use the notation : $O$ : for the normal-ordered form of the operator $O$.

In this work, we construct $\Psi_{\mathrm{ref}}$ using either $f_{1b}$ or $f_{GP}$. The function $f_{1b}$ is the ground-state wave function of the one-boson Hamiltonian as in Refs. [1,2]. The second function is obtained within a mean-field approximation, i.e., $f_{GP}$ is the solution of the Gross-Pitaevskii equation. $f_{1b}$ and $f_{GP}$ are real functions in our work. This choice does not affect the generality of our results, since the ground state of our problem can be described using a real wave function.

To distinguish the IM-SRG method with $f_{1b}$ from IM-SRG with $f_{GP}$, we introduce the notation IM-SRG($f_{1b}$) and IM-SRG($f_{GP}$), respectively[3]. It is worth noting that one can rely on an iterative procedure to find a good reference state, starting from any reasonable initial guess $f_a^{(0)}$. Indeed, IM-SRG($f_a^{(0)}$) may provide a new reference state as $f_a^{(1)} = \sqrt{\rho}$, where $\rho$ is the density obtained from IM-SRG($f_a^{(0)}$). The iterative procedure is continued until $f_a^{(i+1)} \to f_a^{(i)}$, which signals that the results are converged. In addition, this procedure can be used to validate the convergence of our results. We have checked that the results within the zeroth-order iteration (i.e., of IM-SRG($f_a^{(0)}$) with $f_a^{(0)} = f_{1b}, f_{GP}$) are already accurate for the systems discussed here.

## 2.4 Observables

In nuclear physics, the IM-SRG method was used not only to calculate the energy, but also to estimate other observables [5,35]. One of the goals of the present paper is to develop (and test the accuracy of) the IM-SRG method for calculating the density and phase fluctuations of cold Bose gases.

In order to calculate observables other than the energy, the corresponding Hermitian operator $\boldsymbol{O}$ should be transformed together with the Hamiltonian. To this end, we write $\boldsymbol{O}$ in second quantization, normal order it with respect to the reference state $\Psi_{\mathrm{ref}}$, and solve the flow equation

$$\frac{\mathrm{d}\boldsymbol{O}}{\mathrm{d}s} = [\boldsymbol{\eta}, \boldsymbol{O}]. \tag{11}$$

Equations (1) and (11) are solved simultaneously since the generator $\boldsymbol{\eta}$ depends on $\boldsymbol{H}$.

In this work, we focus on calculating one-body observables. The commutator in Eq. (11) leads to two- and higher-order terms for such observables at $s > 0$, which should be truncated according to our scheme. We cannot estimate the associated error using the strategy adopted for the energy (see Appendix A.1), as in general the operators $\boldsymbol{O}$ and $\boldsymbol{H}$ do not commute. Instead we define the "relative truncation error" as

$$\Delta = \frac{\delta e}{e}, \tag{12}$$

where $e$ is the energy calculated using flow equations and $\delta e$ is our estimation of the truncation error for the energy, (see Appendix A.1). We estimate the error due to truncation for $\boldsymbol{O}$ as

$$\delta O \approx \Delta \cdot \langle \boldsymbol{O} \rangle. \tag{13}$$

By comparing to the exact density, we will show below that $\delta O$ can estimate accurately the error of the IM-SRG. However, we do not expect this always to be the case. In general, one cannot infer the accuracy of an observable from the accuracy of the energy using a linear approximation[4], which means that $\delta O$ is no more than a useful phenomenological estimate.

---

[3]Note that this differs from the convention in nuclear physics, where the notation IM-SRG($n$) is used to specify the order $n$ of the truncation scheme for many-body forces.

[4]To illustrate this statement, let us assume that a numerical method produces the following approximation to the ground state: $\psi_0 + \alpha f$, where $\psi_0$ is the exact ground-state wave function, and $f$ is an element of the Hilbert space, which is orthogonal to $\psi_0$, i.e., $\langle \psi_0 | f \rangle = 0$. If the numerical method is accurate, then $\alpha \to 0$. In this case, it is easy to see that the error produced by the numerical method is proportional to $\alpha^2$ for the expectation value of the energy. However, the error for a general observable can be much larger, as it scales as $\alpha$.

## 3   A Bose Gas with a Single Impurity Atom

To illustrate calculations of observables based upon flow equations, we investigate a one-dimensional system of $N$ bosons and a single impurity atom. This system, which is often referred to as the 1D Bose-polaron problem, is one of the simplest models where our approach is useful. The corresponding Hamiltonian in first quantization reads as

$$H = -\frac{\hbar^2}{2m}\frac{\partial^2}{\partial y^2} - \frac{\hbar^2}{2M}\sum_{i=1}^{N}\frac{\partial^2}{\partial x_i^2} + V_{ib}(\{x_i\}) + V_{bb}(\{x_i\}),\tag{14}$$

where $y$ is the position of the impurity, $x_i$ is the position of the $i$th boson, $m$ is the mass of the impurity atom, and $M$ is the mass of a boson. To model atom-atom interactions, we use zero-range potentials [36]:

$$V_{ib}(\{x_i\}) = c\sum_{i=1}^{N}\delta(x_i - y), \qquad V_{bb}(\{x_i\}) = g\sum_{i,j}\delta(x_i - x_j),\tag{15}$$

where $c$ defines the strength of the boson-impurity interactions, and $g$ determines the boson-boson interactions. We consider periodic boundary conditions, i.e., particles are confined to a ring of length $L$, see Fig. 2. The average density of the Bose gas is $\rho = N/L$. We introduce the dimensionless set of parameters:

$$\tilde{x}_i := x_i\rho, \qquad \tilde{y} := y\rho, \qquad \tilde{E} := \frac{M}{\hbar^2\rho^2}E,$$

$$\tilde{m} := \frac{m}{M}, \qquad \tilde{c} := \frac{M}{\hbar^2\rho^2}c, \qquad \gamma := \frac{M}{\hbar^2\rho^2}g,$$

where $E$ is the energy of the system. The parameter $\gamma$ is also known as the Lieb-Liniger parameter. In this new set of units the Hamiltonian $\tilde{H}$ reads as:

$$\tilde{H} = -\frac{1}{2\tilde{m}}\frac{\partial^2}{\partial\tilde{y}^2} - \frac{1}{2}\sum_{i=1}^{N}\frac{\partial^2}{\partial\tilde{x}_i^2} + V_{ib}(\{\tilde{x}_i\}) + V_{bb}(\{\tilde{x}_i\}),$$
$$V_{ib}(\{\tilde{x}_i\}) = \tilde{c}\sum_{i=1}^{N}\delta(\tilde{x}_i - \tilde{y}), \qquad V_{bb}(\{\tilde{x}_i\}) = \gamma\sum_{i,j}\delta(\tilde{x}_i - \tilde{x}_j).\tag{16}$$

In the following we will omit the tilde for better clarity.

We focus on weakly-interacting Bose gases (i.e., $\gamma \ll 1$) that are 'large enough' (i.e., the healing length, $\xi = 1/(\sqrt{\gamma}\rho)$, is smaller than $L$). Furthermore, we assume that the phase coherence length is larger or comparable to $L$, so that the Bose gas is in a (quasi)-condensed state and our reference state is accurate.

In this work, we mainly focus on repulsive boson-impurity interactions ($c > 0$). In Appendix B we present preliminary results for the Bose-gas density and phase fluctuations for attractive interactions. However, further studies are needed for $c < 0$ and will be addressed in a future publication [37]. The self-energy of the impurity for $c > 0$ was already calculated with IM-SRG in Ref. [2]. We are now going to compute other observables using this numerical method. To that end, we transform the Hamiltonian (14) to the system of coordinates co-moving with the impurity, i.e., we introduce a new set of coordinates

$$z_i = N\theta(y - x_i) + x_i - y,\tag{17}$$

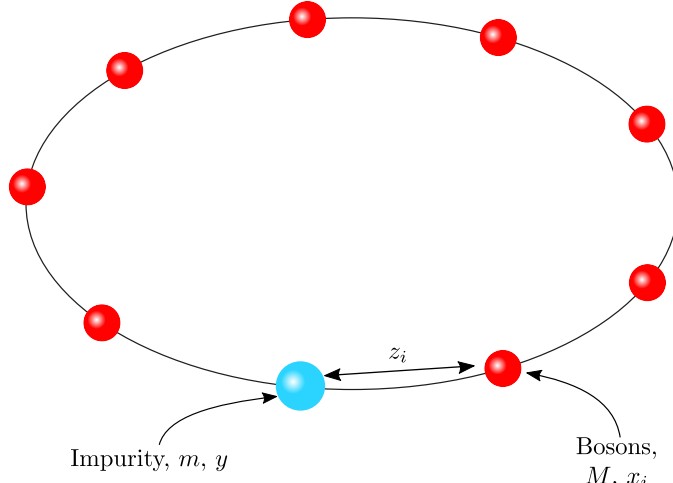

Figure 2: Illustration of the one-dimensional Bose-polaron problem: $N$ bosons and a single impurity on a ring of length $L$. The mass of the impurity atom is $m$ while the mass of a boson is $M$. The coordinates of the impurity and the bosons are $y$ and $\{x_i\}$, respectively. For convenience, the problem is solved using the set of coordinates $\{z_i\}$, which describe the relative distances between the impurity and the bosons.

where $\theta(x)$ is the Heaviside step function. The coordinates $z_i$ allow one to calculate mean-field properties of the Bose-polaron problem in a simple manner, see Refs. [2, 17]. The transformation $\{y, x_i\} \rightarrow \{z_i\}$ is related to the unitary Lee-Low-Pines transformation [38] performed in coordinate space [19, 22]. In the new coordinates, the Hamiltonian (14) is written as

$$\mathcal{H}_P = -\frac{1}{2}\sum_i^N \frac{\partial^2}{\partial z_i^2} - \frac{1}{2m}\left(\sum_i^N \frac{\partial}{\partial z_i}\right)^2 + \frac{iP}{m}\sum_i^N \frac{\partial}{\partial z_i} + \gamma \sum_{i<j}\delta(z_i - z_j) + c\sum_{i=1}^N \delta(z_i), \qquad (18)$$

where $P$ is a quantum number – the total (angular) momentum of the system. We consider the case $P = 0$, as it corresponds to the ground-state manifold. The Hamiltonian $\mathcal{H}_P$ describes a system of $N$ bosons, and can be easily written in the language of second quantization using the annihilation and creation operators, $a_i$ and $a_i^\dagger$, which are defined in the frame co-moving with the impurity.

## 3.1 Reference state

In this section, we introduce the reference states, which are needed to solve the Bose-polaron problem with flow equations. The first reference state is built upon the ground state, $f_{1b}$, of $\mathcal{H}_{P=0}$ for a single boson, i.e., $N = 1$. The corresponding Schrödinger equation reads as

$$-\frac{1}{2\kappa}\frac{\mathrm{d}^2}{\mathrm{d}z^2}f_{1b}(z) + c\delta(z)f_{1b}(z) = \frac{k^2}{2\kappa}f_{1b}(z), \qquad (19)$$

where $\kappa = m/(1 + m)$ is the reduced mass, and $z \in [0, N]$ is the coordinate of a boson. The ground-state solution that satisfies the boundary conditions $[f_{1b}(0) = f_{1b}(N)]$ is $f_{1b} = \mathcal{N}\cos(k(z - N/2))$, where $k \in [\pi/N, 2\pi/N]$ is determined from the equation $2k\tan(kN/2) = 2c\kappa$. The parameter $\mathcal{N}$ is determined from the normalization condition $\int |f_{1b}(z)|^2 \mathrm{d}z = 1$: $\mathcal{N} = \sqrt{2k/(kN + \sin(kN))}$.

The second reference state, $f_{GP}$, solves the Gross-Pitaevskii equation that corresponds to $\mathcal{H}_0$:

$$-\frac{1}{2\kappa}\frac{\mathrm{d}^2 f_{GP}}{\mathrm{d}z^2} + \gamma(N-1)f_{GP}(z)^3 + c\delta(z)f_{GP}(z) = \mu f_{GP}(z),\tag{20}$$

where $\mu$ is the chemical potential. The solution to this equation is given by

$$f_{GP}(z) = \sqrt{\frac{4K(p)^2 p}{\kappa\gamma\delta^2 N^2(N-1)}}\,\mathrm{sn}\left(2K(p)\left[\frac{z}{\delta N} + \frac{1}{2} - \frac{1}{2\delta}\right],p\right),\tag{21}$$

where sn is the sn−Jacobi elliptic function, and $K(p)$ is the complete elliptic integral of the first kind [39]. The parameters $p \in [0,1)$ and $\delta$ are fixed by the boundary conditions due to the delta-function potential $c\delta(z)$ $[f'_{GP}|_{+0} - f'_{GP}|_{-0} = 2\kappa c f_{GP}(0)]$, and by the normalization condition $[\int |f_{GP}(z)|^2 \mathrm{d}z = 1]$. The corresponding chemical potential $\mu$ reads as:

$$\mu = 2\frac{p+1}{\kappa\delta^2 N^2}K(p)^2.\tag{22}$$

The mean-field solution $f_{GP}$ is discussed in more detail in Ref. [2], see also Refs. [19, 22] for the discussion of the thermodynamic limit, and Refs. [40, 41] for the discussion of the limit $c \to \infty$.

## 3.2  Density

In this section, we calculate the density, $\rho(z) = \langle\Phi_{gr}|\boldsymbol{\rho}(z)|\Phi_{gr}\rangle$, of the Bose gas in the frame co-moving with the impurity:

$$\rho(z) = \langle\Phi_{gr}|\sum_{i=1}^{N}\delta(z-z_i)|\Phi_{gr}\rangle,\tag{23}$$

where $\Phi_{gr}$ is the ground state of $\mathcal{H}_0$. $\rho(z)$ should not be confused with the density of the Bose gas without the impurity, $\rho = N/L$. To use the density operator $\boldsymbol{\rho}(z)$ in the flow equation approach (see Sec. 2.4), we write it in second quantization as

$$\boldsymbol{\rho}(z) = \sum_{i,j}\phi_i(z)\phi_j(z)\boldsymbol{a}_i^\dagger\boldsymbol{a}_j,\tag{24}$$

where $\phi_i(z)$ is the $i$th element of the one-body basis employed for writing the Hamiltonian in second quantization. Note that in our implementation the basis $\{\phi_i(z)\}$ depends on the used reference state, see Appendix A.2.

To test flow equations, we first calculate $\rho(z)$ assuming equal masses ($m = M$) and equal interactions ($c = \gamma$). For the ground state, these assumptions turn our system into the exactly solvable Lieb-Liniger model for $N+1$ particles [43] in a ring, since we can no longer distinguish between the impurity and a boson. To calculate the density of the bosons in the frame co-moving with the impurity (see Eq. (23)), we relate it to the pair correlation function of the Lieb-Liniger model, $g_{LL}^{(2)}$:

$$\rho(z) = \frac{N+1}{N}g_{LL}^{(2)}(0,z).\tag{25}$$

To derive this relation, we have used the following definition of $g_{LL}^{(2)}$:

$$g_{LL}^{(2)}(0,z) \equiv N^2\frac{N}{N+1}\int |\Psi_{LL}(y=0,x_1=z,x_2,...,x_N)|^2 \mathrm{d}x_2...\mathrm{d}x_N,\tag{26}$$

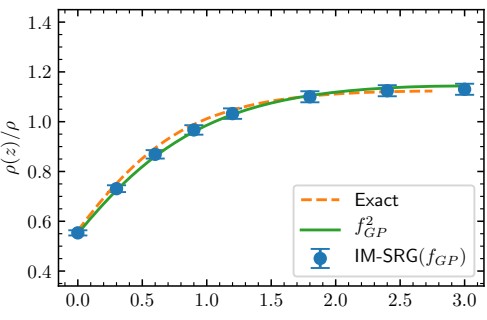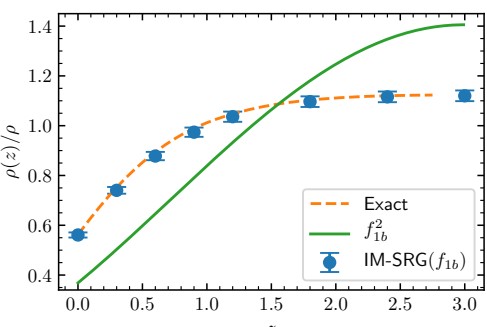

Figure 3: Density of the Bose gas in the frame co-moving with the impurity. The dots are calculated with the flow equations. Left [right] panel shows the results of the IM-SRG($f_{GP}$) [IM-SRG($f_{1b}$)] method. The densities obtained directly from the reference state $f_{GP}$ [$f_{1b}$] are given by the solid green curves. The exact pair-correlation function from Ref. [42] is shown as a dashed orange curve. Our results are presented for $N = 6$ bosons plus a single impurity atom. The interaction strengths are $c = \gamma = 1$, and the masses of impurity and bosons are equal.

where $\Psi_{LL}$ is the ground-state wave function of the Lieb-Liniger model. For $z > 0$, we can write that

$$\Psi_{LL}(y = 0, x_1 = z, x_2, ..., x_N) = \sqrt{\frac{1}{N}} \Phi_{gr}(z, z_2, ..., z_N), \tag{27}$$

which in combination with Eq. (23) leads to Eq. (25).

For certain parameter regimes, the function $g_{LL}^{(2)}$ is known for few-body systems (see, e.g., Ref. [42]), and we use those results to benchmark our findings, see Fig. 3 for $N = 6$. The density calculated using flow equations agrees well with the exact values for all values of $z$. Near the impurity, the density of the bosons is suppressed, since the boson-impurity interaction is repulsive. The presented error bars show the error due to the truncation of the Hilbert space (see, Appendix A.3), and due to the truncation of many-body forces in flow equations, see Eq. (13). For the considered parameters, the latter dominates. All in all, the comparison to the exact results allows us to conclude that our error estimate is accurate.

Figure 3 shows that IM-SRG($f_{1b}$) and IM-SRG($f_{GP}$) agree, which means that both $f_{1b}$ and $f_{GP}$ are suitable reference states for the considered parameters. In our studies, we noticed that the reference state $f_{GP}$ is generally a better choice than $f_{1b}$. In comparison to IM-SRG($f_{1b}$), the scheme IM-SRG($f_{GP}$) allows us to obtain converged results for a larger range of parameters. In particular, IM-SRG($f_{GP}$) is more reliable for large systems, and large boson-boson interactions. Figure 3 explains this observation: The more complicated mean-field function $f_{GP}$ provides a better approximation of the exact density, and, hence, it is easier for the flow equation method to map this reference state onto the real ground state of the system. In what follows, we present our results only for IM-SRG($f_{GP}$).

Finally, we calculate the density for parameters for which the system is no longer exactly solvable. Our goal here is to test the mean-field treatment of $\mathcal{H}_0$. It is already known that this treatment can produce accurate results for the energy of the impurity and the contact parameter [2, 19, 22]. Here, we show that it can also be used to calculate the density, see Fig. 4. As in Fig. 3, the density of the Bose gas is suppressed near the impurity. The length scale that characterizes this suppression is given by $\xi/\sqrt{\kappa}$ [2, 22] (for $\gamma = 0.02$, $\xi\rho \simeq 7.1$ and for $\gamma = 0.1$, $\xi\rho \simeq 3.2$). All in all, we observe that the mean-field approximation describes $\rho(z)$ accurately for all values of the boson-impurity interaction strength $c$, as long as the parameter

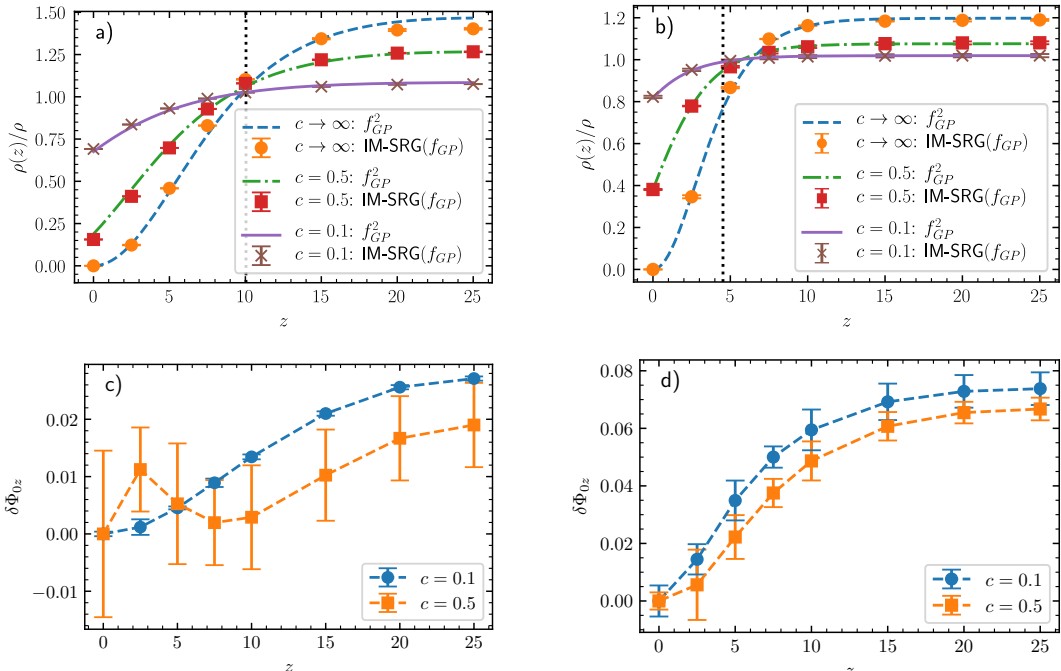

Figure 4: (Upper row): The density of the Bose gas in the frame co-moving with the impurity. Dots are calculated using the IM-SRG($f_{GP}$) method. Mean-field densities are shown using solid, dashed and dot-dashed curves. The dotted vertical line indicates the relevant healing length $\xi/\sqrt{\kappa}$. Results are presented for $N = 50$, $m = 1$ and different impurity-boson interactions $c$ listed in the legend. (Bottom row): Phase fluctuations for the Bose gas in the frame co-moving with the impurity. Dots with error bars are calculated using the IM-SRG($f_{GP}$) method. The dashed curves are added to guide the eye. The parameters $N$ and $m$ are as in the upper row. The boson-boson interaction strength in panels a) and c) is $\gamma = 0.02$, while in b) and d) it is $\gamma = 0.1$.

$\gamma$ is small[5].

We have checked that the mean-field approximation is accurate for up to $\gamma \simeq 0.5$ and $c \to \infty$ by comparing to the Monte-Carlo results presented in Ref. [14]. The comparison of our IM-SRG results to the RG results[6] of Ref. [14] suggests that it is more advantageous to work with a real-space formulation of the Bose-polaron problem. Large values of $c$ require beyond-Fröhlich-polaron treatment of the problem in momentum space, in particular, one should include phonon-phonon interactions. In contrast, in our implementation, already mean-field results are accurate. The accuracy of MFA is probably not surprising after we notice that the (phase) coherence length is larger than the length scale we are interested in. For instance, the phase coherence length for $\gamma = 0.1$ is about $20000\xi$. We illustrate this statement further by calculating phase fluctuations in the next subsection.

---

[5]We expect the mean-field approximation to break down when the boson-boson interaction is increased. With the IM-SRG method we were not able to investigate this, since the truncation error also increases with the boson-boson interaction, e.g., for $N = 15$, $c = 1$ and $\gamma = 1$ the truncation error is $\approx 5\%$ but for $\gamma = 2$ it is already above 10%. These large error bars make it impossible to pinpoint parameters for which the mean-field description starts to fail.

[6]Reference [14] studies large systems with impurities using the (Wilson-type) renormalization group technique in momentum space [44, 45].

### 3.3 Phase fluctuations

Phase fluctuations are strong in the one-dimensional world [33,34,46,47], incapacitating the mean-field treatment. However, as long as one is interested in the physics on the length scales smaller than the coherence length, the mean-field approach can give accurate results. We will now calculate phase fluctuations using flow equations, and explicitly justify the use of the MFA[7]. Another way to validate the mean-field ansatz could be based on calculating the condensate fraction in the considered mesoscopic ensembles. For example, the flow equation approach predicts that the condensate fraction[8] for systems with $N = 50$ and $\gamma \leq 0.1$ is always large ($\sim 95\%$), thus endorsing the use of the mean-field ansatz for these systems. In particular, this large condensate fraction suggests that if fifty bosons can screen the impurity, then the MFA should yield accurate results for the properties of the impurity in the thermodynamic limit. However, the condensate fraction can only be used to provide a phenomenological argument in support of the mean-field approach. In one spatial dimension, the condensate fraction depends on the considered number of particles, and vanishes in the thermodynamic limit. Therefore, we do not discuss it further.

To calculate phase fluctuations, we first compute the one-body density matrix

$$\rho(0,z) \equiv \langle \Phi_{gr} | \boldsymbol{\rho}(0,z) | \Phi_{gr} \rangle = \langle \Phi_{gr} | \sum_{i,j} \phi_i^*(0)\phi_j(z) a_i^\dagger a_j | \Phi_{gr} \rangle \, , \tag{28}$$

and then extract the phase fluctuations $\delta \Phi_{0z}$ using the expression [46]:

$$\rho(0,z) = \sqrt{\rho(0)\rho(z)} \exp\left\{ -\frac{\delta \Phi_{0z}}{2} \right\} . \tag{29}$$

The result is shown in Fig. 4. For $\gamma = 0.02$, phase fluctuations are negligibly small[9]. For $\gamma = 0.1$, phase fluctuations play a more important role, however even then they can be neglected so that $\rho(0,z) \simeq \sqrt{\rho(0)\rho(z)}$. This implies that for these parameters the Bose gas can be described using a mean-field ansatz. One could anticipate that phase fluctuations depend noticeably on the value $c$, which determines the density of bosons and, hence, the effective Lieb-Liniger parameter in the vicinity of the impurity. However, we observe that phase fluctuations depend only weakly on the boson-impurity interaction strength, $c$. Note that in practice it is difficult to extract phase fluctuations according to the definition (29) in systems with $c/\gamma \gg 1$ for which $\rho(0) \to 0$. Therefore we do not compute phase fluctuations in this limit.

### 3.4 Contact parameter

This section focuses on repulsive boson-impurity interactions, and leaves a thorough IM-SRG study of the attractive case for the future [37]. However, we must note that our conclusion that the mean-field approximation describes accurately an impurity in a Bose gas should not be straightforwardly extended to a Bose gas with an attractive impurity. There is an important difference between the cases with $c > 0$ and $c < 0$: The latter supports a formation of tightly bound states at large negative values of $c$. Many-body bound states are highly-correlated and cannot be accurately described by the mean-field ansatz, unlike the opposite case with $c \to \infty$. This implies that the region of applicability for $c > 0$ must be larger (see also the discussion on the effective mass in Ref. [22]). To elaborate slightly more on this difference, we calculate the

---

[7]Phase fluctuations vanish in the MFA.

[8]By condensate fraction, we mean here the expectation value of the operator $a_0^\dagger a_0 / N$.

[9]Note that we expect that the exact curve for $c = 0.5$ in Fig. 4 c) is monotonous. Our calculations of this curve have large error bars, which allow for an apparently non-monotonous behavior.

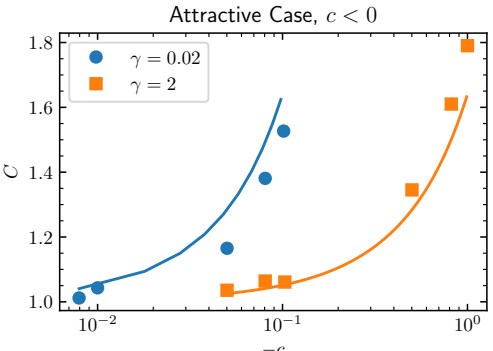
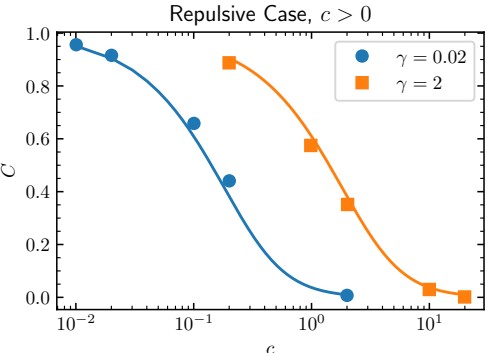

Figure 5: Comparison of the mean-field contact parameter from Eq. (30) to the Quantum Monte-Carlo result [12] for two different boson-boson interaction strengths $\gamma = 0.02$ (blue circles) and $\gamma = 2$ (orange squares). The results are shown as a function of the impurity-boson interaction strength, $c$ with $c < 0$ [$c > 0$] displayed in the left [right] panels, respectively. The solid curves give the mean-field results, while the circles/squares are from Ref. [12]. The masses of the impurity and the bosons are equal ($m = 1$).

density of the Bose gas at the position of the impurity, i.e., the contact parameter, $C = \rho(0)/\rho$, in the mean-field approximation in the thermodynamic limit:

$$C_{MF} = \tanh^{\pm 2}(D), \qquad D = \frac{1}{2}\mathrm{asinh}\left(\frac{2}{c}\sqrt{\frac{\gamma}{\kappa}}\right), \qquad (30)$$

where the positive sign of the exponent is for $c > 0$ (see Ref. [2]) and the negative sign should be taken for $c < 0$. We compare the parameter $C_{MF}$ to the contact parameter calculated using Quantum Monte-Carlo [12] in Fig. 5. For $c > 0$, the agreement between Monte-Carlo and the mean-field approximation is reasonable for all available data points. For attractive interactions, the difference between the results is more noticeable, which implies that the MFA leads to less accurate results for $c < 0$, see also Appendix B, where we present some additional data for the case with attractive interactions. Note in particular Fig. 15, which indicates large phase fluctuations for moderate impurity-boson interactions, in contrast to the repulsive case.

# 4 A Bose Gas with Two Impurity Atoms

In the previous section, we considered a single impurity in a Bose gas, which is the standard starting point in the analysis of systems with impurities. However, the physics of systems with many impurity atoms can be drastically different from that of a system with a single impurity atom, in particular, because the Bose gas mediates interactions between impurities. Therefore, the next step for a reliable description of a (quasispin-)polarized systems must be an assessment of the strength of the induced impurity-impurity interaction potential. One possible way to do this is to consider a Bose gas with two *mobile* impurities, see, e.g., Refs. [20, 25, 48, 49]. We choose another approach. We estimate the induced impurity-impurity interaction using the Born-Oppenheimer approximation [50–52] (see Refs. [53–57] for related studies in three spatial dimensions), i.e., for $m \to \infty$. It is known that the Born-Oppenheimer approximation captures short-range correlations, which define overall properties of impurity-impurity interactions [50, 58]. Long-range impurity-impurity correlations induced by quantum fluctuations are beyond the range of applicability of the Born-Oppenheimer approximation, in particular, since they can depend on the mass of the impurity [21, 27, 59]. These long-range correlations are

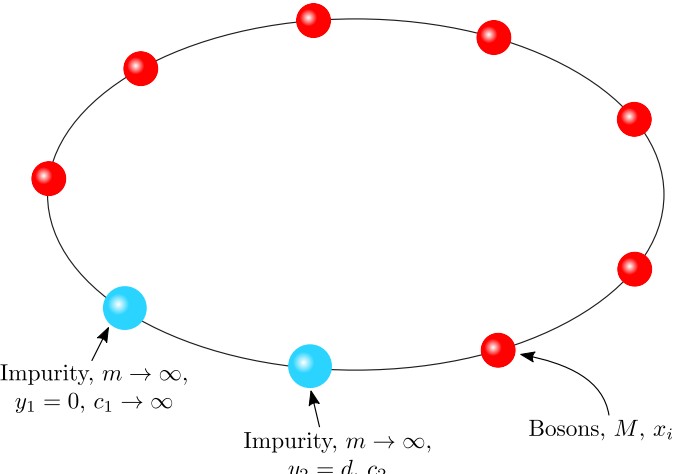

Figure 6: Illustration of a system of $N$ bosons and two static impurities confined to a ring of length $L$. The $i$th boson has the coordinate $x_i$. One impenetrable impurity ($c_1 \to \infty$) is placed at $y_1 = 0$, and another one with the interaction strength $c_2$ is at $y_2 = d$.

not relevant for our discussion – they are weak for the considered parameters regimes, and can be neglected. In particular, these correlations are relevant only at distances $\simeq 5\xi$ (see, e.g., [51]), i.e., they can be extracted only by considering systems that are larger than studied here.

The dimensionless Hamiltonian for two static impurities is written in first quantization as:

$$H_2 = -\frac{1}{2}\sum_{i=1}^{N}\frac{\partial^2}{\partial x_i^2} + \gamma\sum_{i,j}\delta(x_i - x_j) + c_1\sum_{i=1}^{N}\delta(x_i - y_1) + c_2\sum_{i=1}^{N}\delta(x_i - y_2), \qquad (31)$$

where $c_1$ and $c_2$ describe the strength of the impurity-boson interactions, and $y_1$ and $y_2$ are the positions of the impurities. Without loss of generality, we place one impurity at $y_1 = 0$ and the other at $y_2 = d$, see Fig. 6. We assume that the impurity at $y_1$ is impenetrable, i.e., $1/c_1 = 0$. In other words we consider an impurity in a semi-infinite Bose gas [23, 24]. This assumption allows us to simplify the presentation. We will show that for $c_2 > 0$ the impurities attract each other, whereas if $c_2 < 0$ the impurities repel each other. In the former case, the energy is minimized when two impurities are on top of each other, whereas in the latter case the attractive impurity wants to be far from the repulsive one, see also Ref. [23]. We shall consider these cases separately.

The goal of this section is to calculate induced impurity-impurity interactions in the Born-Oppenheimer approximation using the flow equation approach. To this end, below, we compute with IM-SRG the ground-state energy of the Hamiltonian $H_2$. We also show that the induced interactions can be accurately calculated using the mean-field approximation, at least for weak boson-boson interactions. Finally, we estimate when first-order perturbation theory, which is commonly used to estimate impurity-impurity interactions [24, 51], fails.

## 4.1 Repulsive case, $c_2 > 0$

### 4.1.1 Reference state

To use the IM-SRG($f_{GP}$) scheme, we first find the reference state, $f_{GP}$, by solving the Gross-Pitaevskii equation:

$$-\frac{1}{2}\frac{\mathrm{d}^2 f_{GP}}{\mathrm{d}x^2} + \gamma(N-1)f_{GP}(x)^3 + (c_1\delta(x) + c_2\delta(x-d))f_{GP}(x) = \mu f_{GP}(x). \tag{32}$$

The solution to this equation reads

$$f_{GP}(x) = \begin{cases} \sqrt{\dfrac{4K(p_1)^2 p_1}{\gamma \delta_1^2 N^2 (N-1)}}\, \mathrm{sn}\left(2K(p_1)\left[\dfrac{x}{\delta_1 N}\right] + a, p_1\right) & x \in [0,d] \\ \sqrt{\dfrac{4K(p_2)^2 p_2}{\gamma \delta_2^2 N^2 (N-1)}}\, \mathrm{sn}\left(2K(p_2)\left[\dfrac{N-x}{\delta_2 N}\right] + a, p_2\right) & x \in [d,N] \end{cases}, \tag{33}$$

where the parameters $p_1, p_2, \delta_1, \delta_2, a$ are determined by the conditions:

$$\int_0^N f_{GP}^2 \, dx = 1, \tag{34}$$

$$f_{GP}(d^+) = f_{GP}(d^-), \tag{35}$$

$$\frac{p_1+1}{\delta_1^2 N^2}K(p_1) = \frac{p_2+1}{\delta_2^2 N^2}K(p_2), \tag{36}$$

$$\left.\frac{\mathrm{d}f_{GP}}{\mathrm{d}x}\right|_{d^-}^{d^+} = 2c_2 f_{GP}(d), \tag{37}$$

$$\left.\frac{\mathrm{d}f_{GP}}{\mathrm{d}x}\right|_{N^-}^{0^+} = 2c_1 f_{GP}(0). \tag{38}$$

The chemical potential is $\mu = 2\frac{p_1+1}{\delta_1^2 N^2}K(p_1)$. The mean-field solution presented here is valid for all positive values of $c_1$. For the special case $1/c_1 = 0$ that we consider in this section, one should set the parameter $a$ to 0, and solve only the boundary conditions (34)-(37). Once the function $f_{GP}$ is obtained, the mean-field energy of the system is calculated as

$$E_2^{MF} = \mu N - \frac{1}{2}\gamma N(N-1)\int_0^N f_{GP}^4(x)\mathrm{d}x. \tag{39}$$

### 4.1.2 Results

We first calculate the induced interaction potential, $\epsilon_2 = E_2(c_2, d) - E_2(c_2 = 0, d)$, where $E_2$ is the ground-state energy of $H_2$. Our results for this quantity for $N = 60$ particles are presented in Fig. 7. Note that $E_2(c_2 = 0, d) = E_2(c_2, d = 0)$, hence $\epsilon_2(d = 0) = 0$. The potential $\epsilon_2$ is attractive, because the boson-impurity repulsion is minimized when the two impurities are on top of each other. The considered system is not yet in the thermodynamic limit (see the next subsection), but it is large enough to see the saturation of the impurity-impurity interaction at large values of $d$, i.e., far from the impenetrable impurity.

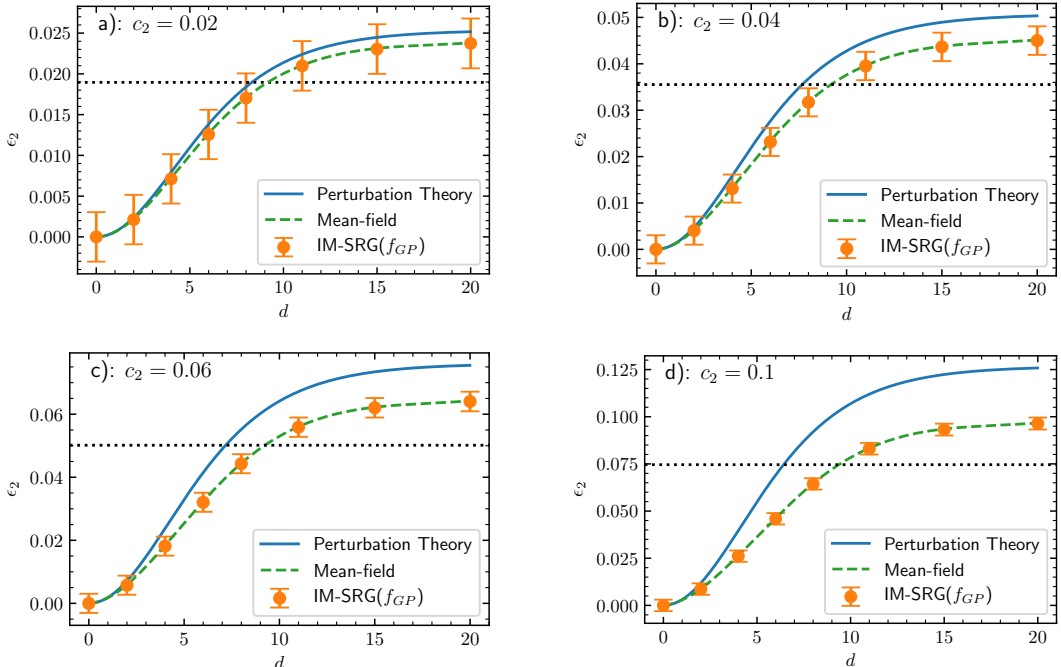

Figure 7: The impurity-impurity interaction induced by the Bose gas for repulsive boson-impurity interactions, $c_2 > 0$. Dots with error bars represent the IM-SRG($f_{GP}$) results. The solid (blue) curves are calculated using perturbation theory, and the dashed (green) curves show the mean-field energy. The data are for $N = 60$, $\gamma = 0.02$ and four different values of the boson-impurity interaction: a) $c_2 = 0.02$, b) $c_2 = 0.04$, c) $c_2 = 0.06$, and d) $c_2 = 0.1$. For comparison, the dotted lines give the self-energies of a single impurity, $E_2(c_1 = 0, c_2) - E_2(c_1 = 0, c_2 = 0)$, for those parameters.

In Fig. 7, we compare the results obtained via IM-SRG($f_{GP}$) and perturbation theory. The latter assumes that the second impurity does not affect the density of the Bose gas, and therefore $\epsilon_2 = c_2 n(d)$, where $n(d)$ is the density of the Bose gas for $c_2 = 0$. This is the common assumption for estimating the induced interaction [23, 24, 51]. The perturbation theory leads to the Yukawa-type potential when both impurities are weakly interacting [25, 50, 51, 58]. Hence, our results indicate the limits of applicability of that standard potential. The figure also presents the mean-field approximation, which uses Eq. (39) to estimate $\epsilon_2$. We conclude that perturbation theory can be used if $c_2 < \gamma$. However, it fails already for $c_2 \gtrsim \gamma$, and more involved calculations are required to find the induced potential in this regime. Perturbation theory implies much stronger impurity-impurity interactions. Its use will lead to wrong predictions for a number of experimentally relevant observables, such as the limits of stability of the polaronic gas [60]. For all considered parameters, the mean-field approximation agrees with the IM-SRG($f_{GP}$) method. This implies that the MFA can be used to calculate the induced potential beyond first order perturbation theory. Finally, we note that far from the impenetrable impurity, the energy $\epsilon_2$ does not approach the self-energy of a single impurity, $E_2(c_1 = 0, c_2) - E_2(c_1 = 0, c_2 = 0)$, (dotted lines in Fig. 7)[10]. This is a finite-size effect. Far from the impenetrable impurity, the density of the Bose gas in a finite system is larger than $\rho$, see Fig. 4, which leads to the difference at $d = 20$ between the dotted lines and dots in Fig. 7.

We also employ the IM-SRG($f_{GP}$) to calculate the density and phase fluctuations of the Bose

---

[10]Note that by definition the self-energy of a single impurity does not depend on $d$.

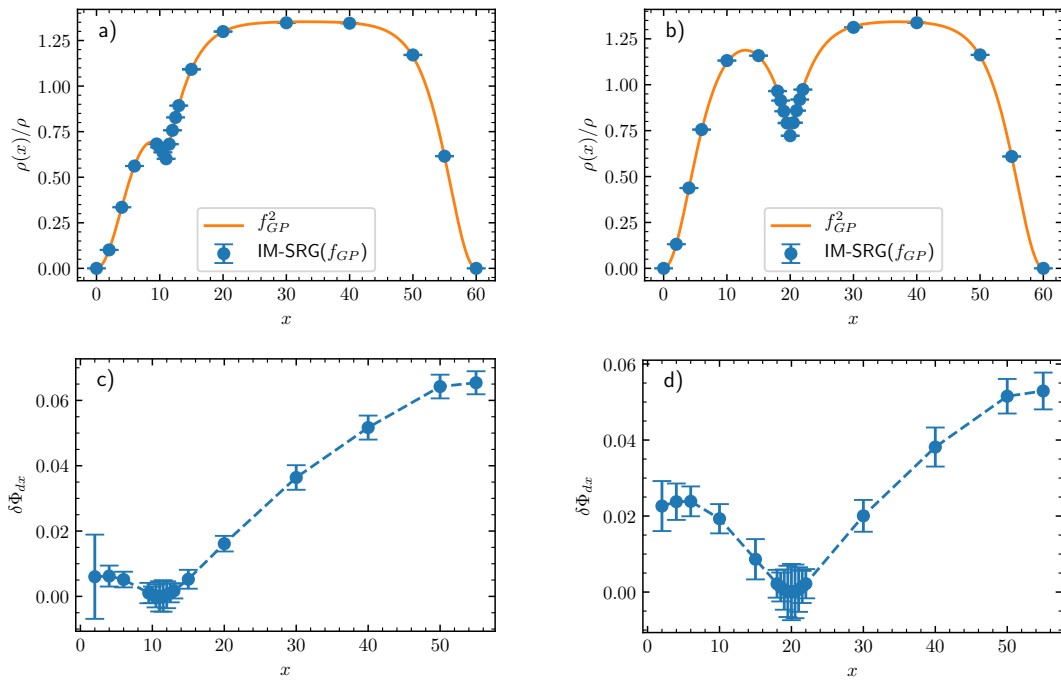

Figure 8: (Upper row): The density of the Bose gas for $N = 60$, $\gamma = 0.02$, $c_2 = 0.1$ and two values of the distance between the impurities: a) $d = 11$ and b) $d = 20$. Dots show the results of the IM-SRG($f_{GP}$). For comparison, the solid curve gives the mean-field density. (Bottom row): Phase fluctuations of the Bose gas for $N = 60$, $\gamma = 0.02$, $c_2 = 0.1$ and two values of the distance between the impurities: c) $d = 11$ and d) $d = 20$. The dots are calculated using the IM-SRG($f_{GP}$).

gas for $N = 60$, $\gamma = 0.02$, $c_2 = 0.1$ and $d = 11, 20$. The results are presented in Fig. 8. The density vanishes at $x = 0$ and $x = 60$ due to the impenetrable impurity and periodic boundary conditions. The Bose gas is strongly affected by the second impurity, which is located at $x = d$. This explains the discrepancy between perturbation theory and the flow equation approach in Fig. 7. All in all, the IM-SRG($f_{GP}$) results for $\rho(x)$ agree well with the mean-field prediction.

Figures 8 c) and d) show phase fluctuations. Here, we choose the position of the second impurity, $d$, as the reference point. As for a single impurity in a ring, phase fluctuations increase far from $d$ where the impenetrable impurity is located[11] (see the discussion at the end of Sec. 3.3). Therefore, we do not show phase fluctuations in the vicinity of those points. The maximum value of $\delta\Phi_{dx}$ is small (in agreement with the results of the previous section), and does not depend strongly on $d$. A condensate fraction for the considered system is about $\sim 95\%$. Our findings presented in this subsection imply that the physics of two strongly interacting impurities in a weakly-interacting Bose gas can be conveniently studied using the mean-field approximation.

## 4.2 Approaching the thermodynamic limit

In this subsection, we study the transition to the thermodynamic limit, i.e., we increase $N$ and $L$, while keeping the density constant, $\rho = N/L$. The key question here is how many bosons are needed to simulate the infinite system. For a single impurity in a weakly-interacting Bose gas, this was briefly considered in Ref. [2]. Here, we discuss what happens for two impurities. The IM-SRG results for $c_2 = 0.1$ and $c_2 = 0.02$, $\gamma = 0.02$ and different numbers of particles are

---

[11]Note, that the density vanishes at $x = 0$ and $x = N$.

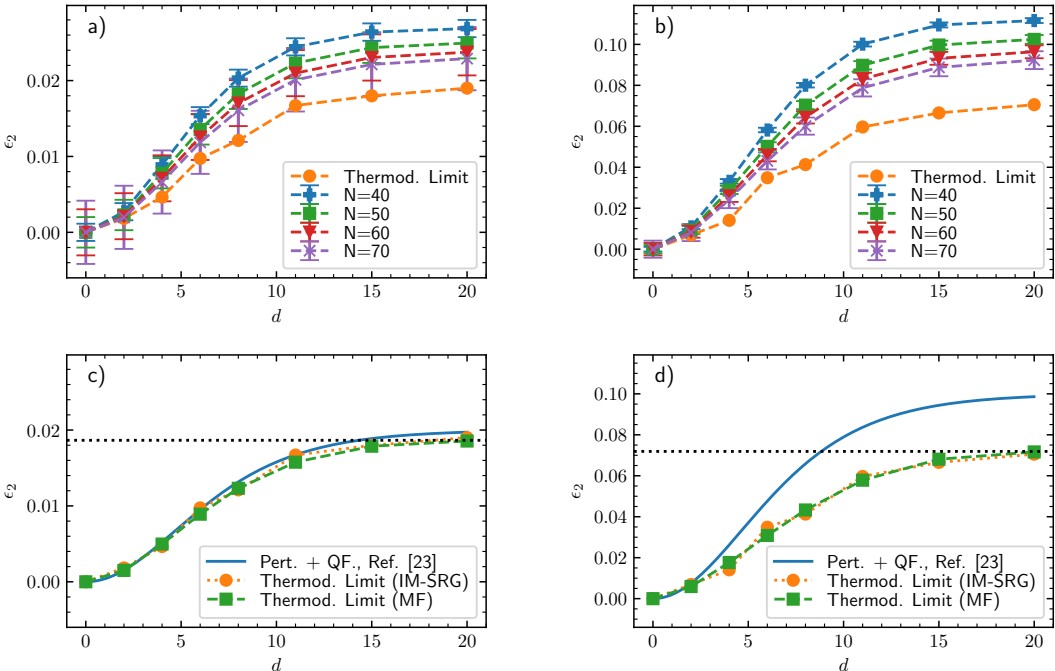

Figure 9: Interaction potential for two repulsive impurities induced by the Bose gas for $\gamma = 0.02$: panel a) shows $c_2 = 0.02$ and panel b) shows $c_2 = 0.1$. The curves display different values of $N$ listed in the insets. We also show our prediction for the induced interaction potential in the thermodynamic limit. No error bars are shown for this case, since we cannot estimate them reliably, see text. In the lower panels, we compare our prediction for the thermodynamic limit (orange squares) with the corresponding result based upon perturbation theory, which includes quantum fluctuations [23] (solid blue curve), and the mean-field result (green dots). Panels c) and d) are for $c_2 = 0.02$ and $c_2 = 0.1$, respectively. The dotted lines in c) and d) show the self-energies of a single impurity, $E_2(c_1 = 0, c_2) - E_2(c_1 = 0, c_2 = 0)$ in the thermodynamic limit while the dashed curves are added to guide the eye.

shown in Fig. 9 a) and b). For the considered values of $N$, the induced interaction is far from the thermodynamic limit, i.e., it changes with the numbers of particles. The thermodynamic limit is reached for system sizes which are beyond the flow equation approach. Still, we can use IM-SRG to predict the induced potential in the thermodynamic limit by fitting the IM-SRG energies to the function $C_1 + \frac{C_2}{N^{C_3}}$, where $C_1, C_2$ and $C_3$ are fitting parameters. The parameter $C_1$ defines the potential in the thermodynamic limit. The fitting parameters $C_2$ and $C_3$ have no direct physical interpretation[12]. In Figs. 9 a) and b), we present the value of $C_1$. The truncation error in the IM-SRG method grows rapidly with the number of particles. This rapid growth rules out a reliable extrapolation of the error bars to the thermodynamic limit. Therefore, we give no estimate for the accuracy of $C_1$, which leads to an apparently oscillating character of the potential in the thermodynamic limit. We expect that the exact potential is a monotonically increasing function of the distance between the impurities, $d$, for the considered values of $d$.

In Figs. 9 c) and d), we compare our estimate for the potential in the thermodynamic limit with the result based upon perturbation theory, which includes quantum fluctuations [23, 24]. For weak boson-impurity interactions, $c_2 = \gamma$, both curves agree well. For larger interactions, however, the curves deviate. In this case, the density of bosons is strongly influenced by the

---

[12]The best fits to the data have the parameter $C_3$ in between 1 and 2, depending on the value of $d$.

second impurity (see Fig. 8), and, therefore, perturbation theory is not a valid approximation. We draw two conclusions from Fig. 9. First, high compressibility of a weakly-interacting Bose gas leads to a large number of particles needed to reach the thermodynamic limit. This should be contrasted with systems of strongly interacting bosons or fermions, for which a handful of particles can screen the impurity [61], for more detail, see Refs. [62, 63] and references therein. Here we find that for $\gamma = 0.02$, one needs more than 100 particles to capture the effective short-range interaction between two static impurities in the thermodynamic limit. This implies that any study that aims to relate measurements in current cold-atom set-ups with small number of particles to the thermodynamic limit must provide an estimation of finite-size effects. This is especially important for any prospective experimental study of induced long-range interactions. Second, calculations beyond first-order perturbation theory are required to estimate induced impurity-impurity interactions also in the thermodynamic limit. As we show here, these calculations can be based upon the mean-field approximation.

## 4.3   Attractive case, $c_2 < 0$

In this section, we consider attractive boson-impurity interactions, $c_2 < 0$. This case is more complicated because all bosons are bound to the impurity if $\gamma N \lesssim 2|c_2|$ in the limit $L \to \infty$ (assuming that $N$ is fixed), see [37, 64]. Therefore, to obtain a meaningful estimation for the induced interactions, we must consider $N \gg 2|c_2|/\gamma$.

### 4.3.1   Reference state

For attractive interactions, we do not obtain the reference state $f_{GP}$ from the Gross-Pitaevskii equation. Instead, we choose the reference state as

$$f_{GP}(x) = \mathcal{N} f_{one-rep}(x) f_{one-attr}(x), \tag{40}$$

where $f_{one-rep}$ is the mean-field solution for a Bose gas with one repulsive impurity, and $f_{one-attr}$ is the mean-field solution for a Bose gas with a single attractive impurity. Therefore, the function $f_{GP}$ in Eq. (40) is the full mean-field solution for two impurities in the limit of large separation between the impurities, i.e., $d \gg 1$. Otherwise, the function $f_{GP}$ is an approximation to the solution of the Gross-Pitaevskii equation. We observe that $f_{GP}$ is an accurate approximation, see, in particular, Fig. 11, where $f_{GP}$ is plotted together with the density of the bosons calculated using flow equations.

The function $f_{one-rep}$ is given by (see Sec. 3.1)

$$f_{one-rep}(x) = \sqrt{\frac{4K(p_1)^2 p_1}{\gamma \delta_1^2 N^2(N-1)}} \, \text{sn}\left(2K(p_1)\left[\frac{x}{\delta_1 N} + \frac{1}{2} - \frac{1}{2\delta_1}\right], p_1\right), \tag{41}$$

where $\delta_1 = 1$, since we work with an impenetrable impurity, $c_1 \to \infty$. The function $f_{one-attr}$ reads as [37]

$$f_{one-attr}(x) = \sqrt{\frac{4K(p_2)^2}{\gamma \delta_2^2 N^2(N-1)}} \, \text{ns}\left(2K(p_2)\left[\frac{x}{\delta_2 N} + \frac{1}{2} - \frac{1}{2\delta_2}\right], p_2\right). \tag{42}$$

The parameter $\mathcal{N}$ is given by the normalization condition:

$$\int f_{GP}(x)^2 \mathrm{d}x = 1. \tag{43}$$

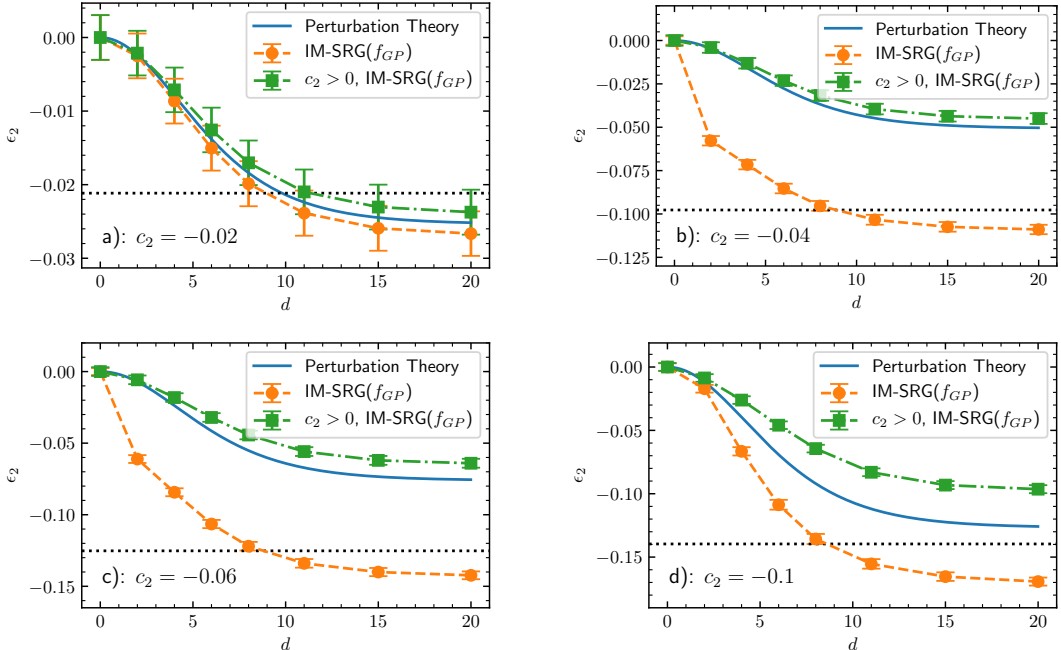

Figure 10: Induced impurity-impurity interaction for a repulsive and an attractive impurity, $c_2 < 0$, $N = 60$, and $\gamma = 0.02$. The four panels are for different values of $c_2$: a) $c_2 = -0.02$, b) $c_2 = -0.04$, c) $c_2 = -0.06$, and d) $c_2 = -0.1$. The circles with error bars show the result of the IM-SRG($f_{GP}$) calculation. The solid curve gives the result of first-order perturbation theory. For comparison, we also show the IM-SRG result for $c_2 > 0$ (squares) times (-1). The dashed and dot-dashed curves are added to guide the eye while the dotted lines show the self-energies of a single impurity, $E_2(c_1 = 0, c_2) - E_2(c_1 = 0, c_2 = 0)$.

### 4.3.2 Results

We compare the induced interaction potential obtained with the IM-SRG($f_{GP}$) to the one obtained using first-order perturbation theory, see Fig. 10. In contrast to the case of $c_2 > 0$, now the induced impurity-impurity potential is repulsive. The attractive impurity maximizes its energy by being far from the hole in the density of bosons created by the repulsive impurity. In agreement with the case $c_2 > 0$, perturbation theory fails to describe impurity-impurity interactions when $|c_2| \gtrsim \gamma$. The important difference is that now perturbation theory leads to a weaker induced interaction in comparison to the IM-SRG result.

As in the previous case, perturbation theory fails because the density of the Bose gas is strongly modified in the vicinity of the impurity for $|c_2| \gtrsim \gamma$. To illustrate a strong modification of the density, we calculate $\rho(x)$, and phase fluctuations of the Bose gas for $N = 60$, $\gamma = 0.02$, $c_2 = -0.1$ and $d = 11, 20$ within the IM-SRG($f_{GP}$), see Fig. 11.

Figure 11 demonstrates that the reference state $f_{GP}$ gives an accurate approximation to the exact density of the Bose gas. Phase fluctuations are small, and we have checked that the condensate fraction for these parameters is above 95%. Therefore, our conclusion for the considered parameters is similar to the case with $c_2 > 0$: The mean-field approach can be used to describe the impurity-impurity mediated interactions as long as the boson-boson interaction is weak. Although, we do not demonstrate this here, we expect that $f_{GP}$ and the mean-field approach in general to be less accurate for one attractive impurity in comparison to one repulsive impurity when $|c_2| \gg \gamma$. This expectation is based on our analysis of the contact parameter, see Fig. 5, as well as on the results presented in Appendix B.

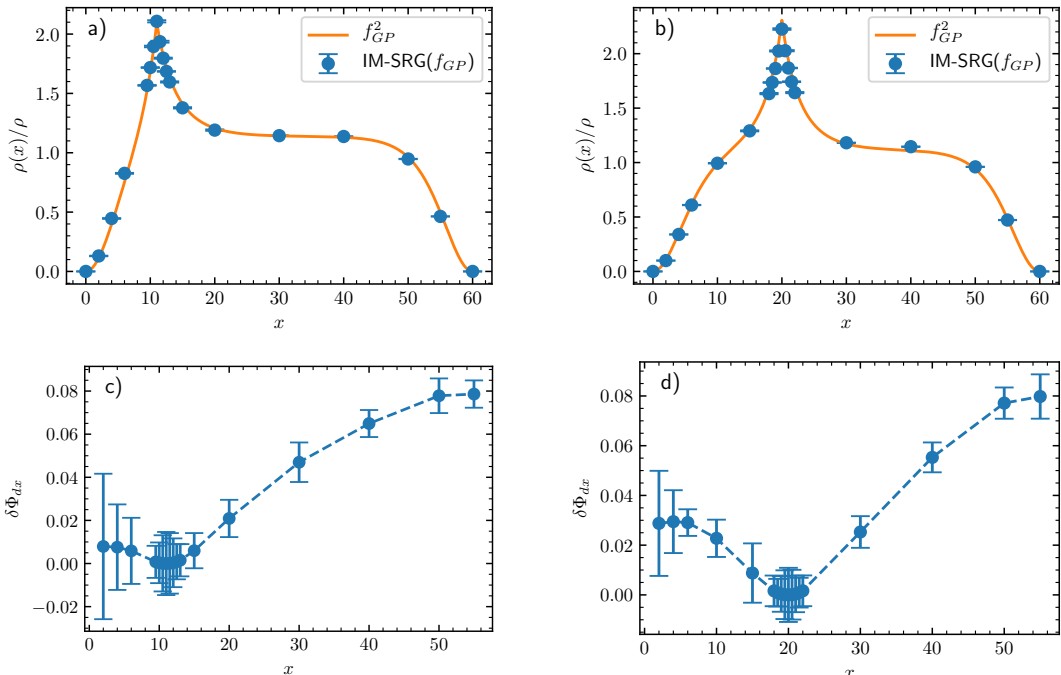

Figure 11: (Upper Row): The density of the Bose gas for $N = 60$, $\gamma = 0.02$, $c_2 = -0.1$ and two different separations of the impurities: a) $d = 11$ and b) $d = 20$. We show the density calculated using flow equations (dots) together with the density calculated using the reference state, $f_{GP}$ (solid curve). (Bottom Row): Phase fluctuations of the Bose gas for $N = 60$, $\gamma = 0.02$, $c_2 = -0.1$ and the separations c) $d = 11$ and d) $d = 20$. The dots show the IM-SRG($f_{GP}$) results. The dashed curves are added to guide the eye.

## 4.4 Approaching the thermodynamic limit

Finally, we calculate the induced interactions for different values of $N$, in order to understand the few- to many-body transition in this system. We illustrate our findings in Fig. 12 for $\gamma = 0.02$ and $c_2 = -0.02$, $c_2 = -0.1$. Like in the case with $c_2 > 0$, we fit the energy to estimate the induced impurity-impurity potential in the thermodynamic limit, and compare the estimate to the result of Ref. [23]. Just like in the case with $c_2 > 0$, we see that perturbation theory is accurate for weak boson-impurity interactions but fails to describe stronger interactions.

## 5 Summary and Outlook

The paper explores the possibility to calculate observables for degenerate Bose gases using the flow equation approach. For illustration purposes, the focus is on a one-dimensional Bose gas with one and two impurity atoms. The considered system allows us to benchmark the IM-SRG results against the existing exact data based upon the Bethe ansatz, and to study in detail the Bose-polaron problem and polaron-polaron interactions, topics of current theoretical interest.

In the single impurity case, we consider repulsive boson-impurity interactions ($c > 0$). It turns out that the density of the Bose gas for the repulsive Bose-polaron problem can be calculated accurately using the mean-field approximation in the coordinate frame, which is 'co-moving' with the impurity. To explain the validity of the mean-field approach, we show that the condensate fraction is large and phase fluctuations are small for the considered mesoscopic

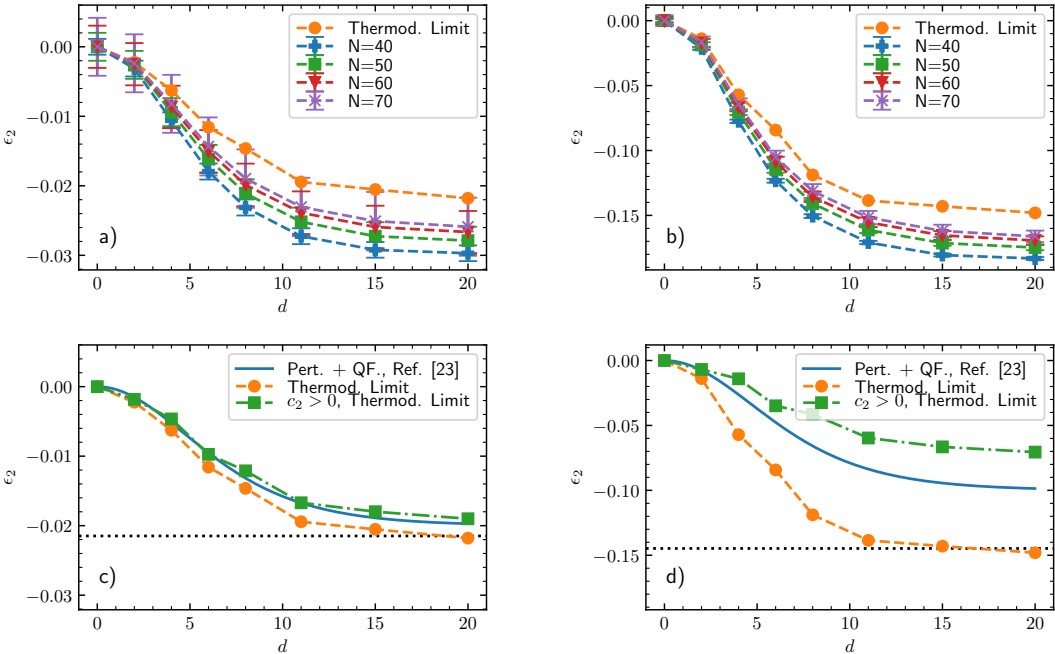

Figure 12: Interaction potential between two impurities induced by the Bose gas for $\gamma = 0.02$ in the attractive case, $c_2 < 0$, with two different strengths a) $c_2 = -0.02$ and b) $c_2 = -0.1$. The curves show different values of $N$ listed in the insets. We also give our prediction for the induced interaction potential in the thermodynamic limit. No error bars are shown for this case, since we cannot estimate them reliably, see text. In the lower panels, we display our prediction for the values in the thermodynamic limit (orange dots) and the corresponding result based upon perturbation theory, which includes quantum fluctuations [23] (solid blue curve). Panels c) and d) are for $c_2 = -0.02$ and $c_2 = -0.1$, respectively. For comparison, we also show the negative IM-SRG result for $c_2 > 0$ (green squares). The dashed curves are plotted to guide the eye while the dotted lines in c) and d) give the self-energy of a single impurity $E_2(c_1 = 0, c_2) - E_2(c_1 = 0, c_2 = 0)$, in the thermodynamic limit.

ensembles ($N < 100$), and weak boson-boson interactions. For attractive interactions ($c < 0$), the possibility of deep impurity-boson bound states complicates the analysis. This issue will be addressed in a future publication [37].

For two impurities, we calculate induced impurity-impurity interactions in the Born-Oppenheimer approximation. For simplicity, we assume that one impurity is impenetrable. The other one either attracts or repels bosons. The former case leads to repulsive, and the latter one to attractive induced interactions. We find that the mean-field approximation describes accurately these interactions, and can be used in straightforward calculations based upon the well-studied Gross-Pitaevskii equation. We also show that first order perturbation theory is valid when boson-impurity interaction is smaller or equal to the boson-boson interaction, however, fails otherwise. Finally, we discuss the few- to many-body transition, and show the importance of finite-size effects for impurities in weakly-interacting Bose gases.

Our results show that the mean-field approach is a robust tool to study weakly-interacting Bose gases with impurities. In the other limit of strongly interacting Bose gases, one can study the self-energy and the density of an impurity using tools developed for Fermi gases, e.g., based upon variational wave functions [65] or the Bethe ansatz [66]. In the future, it might be interesting to investigate the transition between these two limits, and the evolution

of the Bose polaron into the impurity in a Tonks-Girardeau gas. A modification of the flow equation approach with two reference states might be useful to study this transition. One could compare the obtained IM-SRG results to those calculated using beyond-Gross-Pitaevskii effective theories (as, e.g., introduced in Ref. [67]).

The present work paves the way for IM-SRG studies of Bose gases in higher spatial dimensions. A starting point for such an extension might be a study of dilute bosonic droplets, for which a number of exciting analytical predictions exist [68, 69]. It should also be possible to investigate two- and three-dimensional systems with impurities. The relevant mean-field solutions can be found in the literature [70–73], giving reference states for flow equations. A modification of the IM-SRG, which takes into account Hilbert space associated with an impurity can allow one to study composite impurities in Bose gases, and corresponding quasiparticles, in particular, angulons [74, 75].

*Note added after ArXiv submission:* After submission of this manuscript, we learned of recent works [76, 77], where the accuracy of the mean-field approximation to polaron-polaron interactions is also discussed.

**Acknowledgments and Funding information**   We thank Matthias Heinz and Volker Karle for helpful comments on the manuscript; Zoran Ristivojevic for useful correspondence regarding mean-field calculations of induced impurity-impurity interactions; Fabian Grusdt for sharing with us the data for the densities presented in Ref. [14]. This work has received funding from the DFG Project No. 413495248 [VO 2437/1-1] (F. B., H.-W. H., A. G. V.) and European Union's Horizon 2020 research and innovation programme under the Marie Skłodowska-Curie Grant Agreement No. 754411 (A. G. V.). M. L. acknowledges support by the European Research Council (ERC) Starting Grant No. 801770 (ANGULON). H.-W.H. thanks the ECT* for hospitality during the workshop "Universal physics in Many-Body Quantum Systems – From Atoms to Quarks". This infrastructure is part of a project that has received funding from the European Union's Horizon 2020 research and innovation programme under grant agreement No 824093. H.-W. H. was supported by the Deutsche Forschungsgemeinschaft (DFG, German Research Foundation) - Project-ID 279384907 - SFB 1245.

# A   Details on the Method

Below, we explain the IM-SRG method in more detail. We start by presenting the form of the flow equation (1) after our truncation of many-body terms, and discuss our estimate of the truncation error. After that, we explain further technical details of our implementation of the IM-SRG method. First, we discuss the one-body basis, which is used to represent the Hamiltonian in second quantization. Then, we discuss the truncation of the Hilbert space.

In the Appendix, the Einstein summation rule is implied, when Latin indices are used. There is no summation for Greek indices.

## A.1   Flow equation

Our flow equation reads as

$$\frac{\mathrm{d}H}{\mathrm{d}s} = [\eta, H],\tag{44}$$

where the generator $\eta$ is written as

$$\eta(s) = \xi_{ij}(s) : a_i^\dagger a_j : + \frac{1}{2}\eta_{ijkl}(s) : a_i^\dagger a_j^\dagger a_k a_l : .\tag{45}$$

The matrices $\xi_{ij}$ and $\eta_{ijkl}$ must be chosen such that the couplings to the ground state - the matrix elements $f_{i0}$ and $\Gamma_{ij00}$ - vanish (see Fig. 1). Note that the commutator in the right-hand-side of Eq. (1) induces many-body terms:

$$[: a_i^\dagger a_j^\dagger a_k a_l :,: a_a^\dagger a_b^\dagger a_c a_d :] = a_i^\dagger a_j^\dagger a_b^\dagger a_k a_c a_d + \dots . \tag{46}$$

This three-body operator induces a three-body operator in the Hamiltonian at $s > 0$. Since all couplings from the ground state need to vanish, we would now need a three-body operator in the generator as well, which would in turn generate a four-body operator and so on. It is therefore impossible to treat the flow equation exactly and we need to truncate many-body terms. We choose to truncate at the two-body level, keeping only the terms from the three-body operator that contain at least one $a_0^\dagger a_0$ operator (see Ref. [1] for a more detailed discussion). We neglect the remaining pieces (called $W$), see also Fig. 1, which illustrates the used truncation scheme.

Upon truncation, we derive a closed system of coupled differential equations (note that Ref. [1] has typos in this system of equation, which we correct here):

$$\frac{d\epsilon}{ds} = S_{00} + (N-1)\left(\frac{1}{2}S_{00ii00} - S_{000000}\right), \tag{47}$$

$$\begin{aligned}
\frac{df_{\alpha_1\alpha_2}(s)}{ds} &= -(N-1)^2(S_{0\alpha_100\alpha_20} + S_{0\alpha_1\alpha_2000} + S_{000\alpha_1\alpha_20}) + (N-1)S_{\alpha_10ii0\alpha_2} \\
&\quad + (N-1)S_{0\alpha_1\alpha_20} + \frac{(N-1)(N-2)}{2}(S_{00\alpha_2\alpha_100} + S_{0\alpha_100\alpha_20}D_{\alpha_2}D_{\alpha_1}) \\
&\quad + \frac{(N-1)(N-2)}{2}(S_{0\alpha_1\alpha_2000}D_{\alpha_1} + S_{000\alpha_1\alpha_20}D_{\alpha_2}) + S_{\alpha_1\alpha_2},
\end{aligned} \tag{48}$$

$$\begin{aligned}
\frac{d\Gamma_{\alpha_1\alpha_2\alpha_3\alpha_4}(s)}{ds} &= \frac{(1 + P_{\alpha_1\alpha_2})(1 + P_{\alpha_3\alpha_4})}{2}\Big(S_{\alpha_1\alpha_2\alpha_3\alpha_4} \\
&\quad - (N-1)(S_{\alpha_1\alpha_2\alpha_300\alpha_4} + S_{\alpha_100\alpha_2\alpha_3\alpha_4}) + \frac{1}{2}S_{\alpha_1\alpha_2ii\alpha_3\alpha_4} \\
&\quad + (N-2)D_{\alpha_1}D_{\alpha_4}S_{0\alpha_1\alpha_3\alpha_2\alpha_40} + (N-2)I_{\alpha_1\alpha_2}D_{\alpha_4}S_{\alpha_1\alpha_2\alpha_300\alpha_4} \\
&\quad + (N-2)D_{\alpha_1}I_{\alpha_3\alpha_4}S_{\alpha_100\alpha_2\alpha_3\alpha_4} + (N-2)I_{\alpha_1\alpha_2}I_{\alpha_3\alpha_4}S_{\alpha_1\alpha_200\alpha_3\alpha_4}\Big),
\end{aligned} \tag{49}$$

with $D_{\alpha_1} = 2 - \delta_{\alpha_10}$, $I_{\alpha_1\alpha_2} = 1 + \delta_{\alpha_10}\delta_{\alpha_20} - 2\delta_{\alpha_20}$, and

$$\begin{aligned}
S^{(1)}_{\alpha_1\alpha_2} &= \xi_{\alpha_1i}f_{i\alpha_2} - f_{\alpha_1i}\xi_{i\alpha_2}, \\
S^{(2)}_{\alpha_1\alpha_2\alpha_3\alpha_4} &= \xi_{\alpha_1i}\Gamma_{i\alpha_2\alpha_3\alpha_4} - \Gamma_{\alpha_1\alpha_2\alpha_3i}\xi_{i\alpha_4} + \eta_{\alpha_1\alpha_2\alpha_3i}f_{i\alpha_4} - f_{\alpha_1i}\eta_{i\alpha_2\alpha_3\alpha_4}, \\
S_{\alpha_1\alpha_2\alpha_3\alpha_4\alpha_5\alpha_6} &= \eta_{\alpha_1\alpha_2\alpha_3i}\Gamma_{i\alpha_4\alpha_5\alpha_6} - \Gamma_{\alpha_1\alpha_2\alpha_3i}\eta_{i\alpha_4\alpha_5\alpha_6}.
\end{aligned} \tag{50}$$

To estimate the truncation error for the ground-state energy we use second-order perturbation theory [1]

$$\delta e \simeq \frac{1}{N}\sum_p \frac{\left(\langle\Phi_p|\int_0^\infty W(s)ds|\Phi_{\text{ref}}\rangle\right)^2}{\langle\Phi_p|H|\Phi_p\rangle - \langle\Phi_{\text{ref}}|H|\Phi_{\text{ref}}\rangle}, \tag{51}$$

where $\Phi_p$ is a state that contains three-body excitations.

The explicit choice of the generator can be justified *a posteriori* if the couplings to the excited states vanish [3]. Our choice of the generator is

$$\eta = f_{i0} : a_i^\dagger a_0 : + \frac{1}{2}\Gamma_{ij00} : a_i^\dagger a_j^\dagger a_0 a_0 : - \text{h.c.}. \tag{52}$$

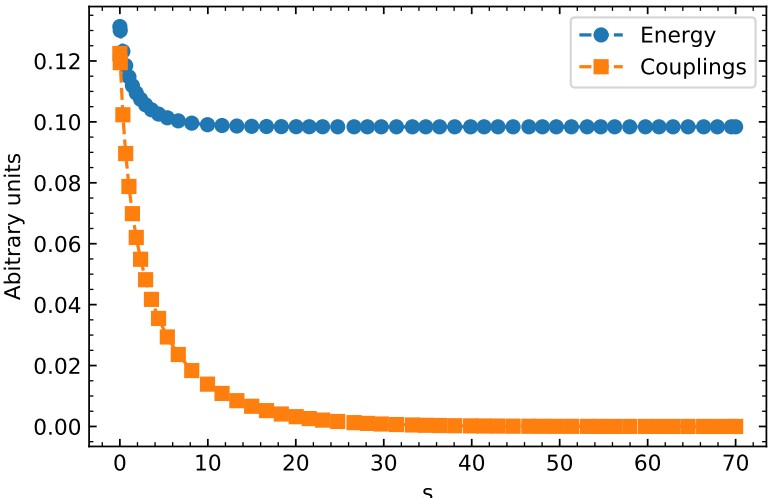

Figure 13: Illustration of a solution of the flow equations (47), (48), (49) as a function of $s$. The ground-state energy $\epsilon N$ is shown using (blue) dots. The sum of the couplings, $\sum_i |f_{i0}| + \sum_{ij} |\Gamma_{ij00}|$, is presented as (orange) squares. The dashed curves are added to guide the eye.

For this generator, the couplings (the matrix elements $f_{i0}$ and $\Gamma_{ij00}$) vanish, and the energy of the ground state, $\epsilon N$, converges, see Fig. 13 for an illustration of a typical convergence pattern in our study. We expect that other standard choices of the generator, see, e.g., Ref. [5], lead to similar results, see also Ref. [1].

## A.2   One-body basis

To write the Hamiltonian in second quantization, $H = A_{ij} a_i^\dagger a_j + \frac{1}{2} B_{ijkl} a_i^\dagger a_j^\dagger a_k a_l$, we need to calculate the matrix elements $A_{ij}$ and $B_{ijkl}$ using some one-body basis. In this section, we discuss the one-body basis used in the present study.

First, we solve the single-boson problem whose eigenstates produce the basis set $\{\phi_i\}$. This set is used as a basis set when we work with the (single-body) reference state $f_{1b} = \phi_0$, where $\phi_0$ is the ground-state of the single-boson problem. The corresponding contractions enjoy the simple form $C_{\alpha_1 \alpha_2} = \delta_{\alpha_1 0} \delta_{\alpha_2 0} N$ and $C_{\alpha_1 \alpha_2 \alpha_3 \alpha_4} = \delta_{\alpha_1 0} \delta_{\alpha_2 0} \delta_{\alpha_3 0} \delta_{\alpha_4 0} N(N-1)$. To keep this simple form of contractions also when we work with the state $f_{GP}$, we construct another one-body basis set: We take $f_{GP}$ as the zeroth element of our basis, and use the Gram-Schmidt process to build an orthogonal basis set from $f_{GP}, \phi_1, \phi_2, \dots$.

## A.3   Truncation of the Hilbert space

We truncate the one-body basis to solve the flow equations numerically. Let us denote the size of the truncated basis by $n$. As argued in Ref. [1], to calculate the energy of the system in the limit $n \to \infty$, one should compute $\epsilon$ for a few values of $n$, and then fit the obtained sequence using the function

$$\epsilon(n) = \epsilon(n \to \infty) + \frac{A}{n^\delta}, \tag{53}$$

where $\epsilon(n \to \infty)$, $A$, $\delta$ are fit parameters. $\epsilon(n \to \infty)$ is the value of the energy in the limit $n \to \infty$. This value is presented in the figures of the main text. In our calculations, we fit results for $n = 13 - 23$. We estimate the uncertainty by the standard deviation error of the fit. We show convergence of other observables in Figs. 14 b), c), and d).

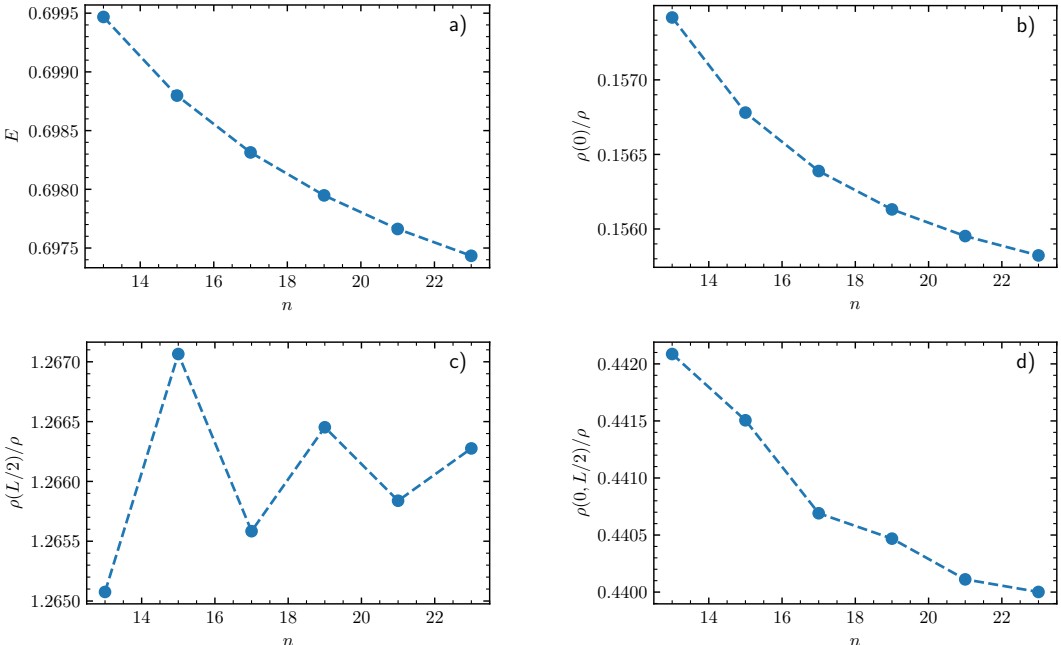

Figure 14: Convergence of observables for a system with one repulsive impurity as a function of the truncation parameter, $n$. The parameters of the system are $N = 50$, $\gamma = 0.02$ and $c = 0.5$. Panel a) shows the energy of the system, panel b) presents the density at $z = 0$, panel c) shows the density at $z = N/2$, and panel d) gives the one-body density matrix at $\rho(0, N/2)$. The dashed curves are added to guide the eye.

Observables that depend on the density of the bosons in the vicinity of the impurity [see Figs. 14 b) and d)] approach the limit $n \to \infty$ in a similar fashion to the energy. Such behavior is the result of a slow convergence of the wave function due to the delta-function potential. To calculate such observables in the limit $n \to \infty$, we use the fitting procedure described above.

Observables that do not depend on the density of bosons in the vicinity of the impurity, e.g., the density of the Bose gas far away from the impurity [see Fig. 14 c)], are virtually converged for $n = 23$. Therefore, to estimate the value of the observable in the limit $n \to \infty$, we take the result for $n = 23$. To estimate the uncertainty of this value, we use the largest difference between the results obtained with $n = 13 - 23$.

## B  An Attractive Impurity in a Bose gas

We leave a rigorous study of a Bose gas with a single attractive impurity to a future publication [37]. However, for the sake of completeness, we briefly discuss here properties of a system defined in Sec. 3 with $c < 0$. In Fig. 15, we present the density of the Bose gas. The figure shows that the mean-field approximation works well for $c = -0.1$, and $c = -0.2$. However, there is a difference between the MFA and IM-SRG results for two largest values of $|c|$. The difference is most noticeable for the density at $z = 0$, which defines the contact parameter (see Sec. 3).

Finally, we calculate phase fluctuations, see Fig. 15 e). The figure demonstrated significant phase fluctuations for $c \simeq 0.25$. This increase signals that the Bose gas and the impurity form a many-body bound state. Large phase fluctuations also suggest that the mean-field ansatz is no longer a valid approximation.

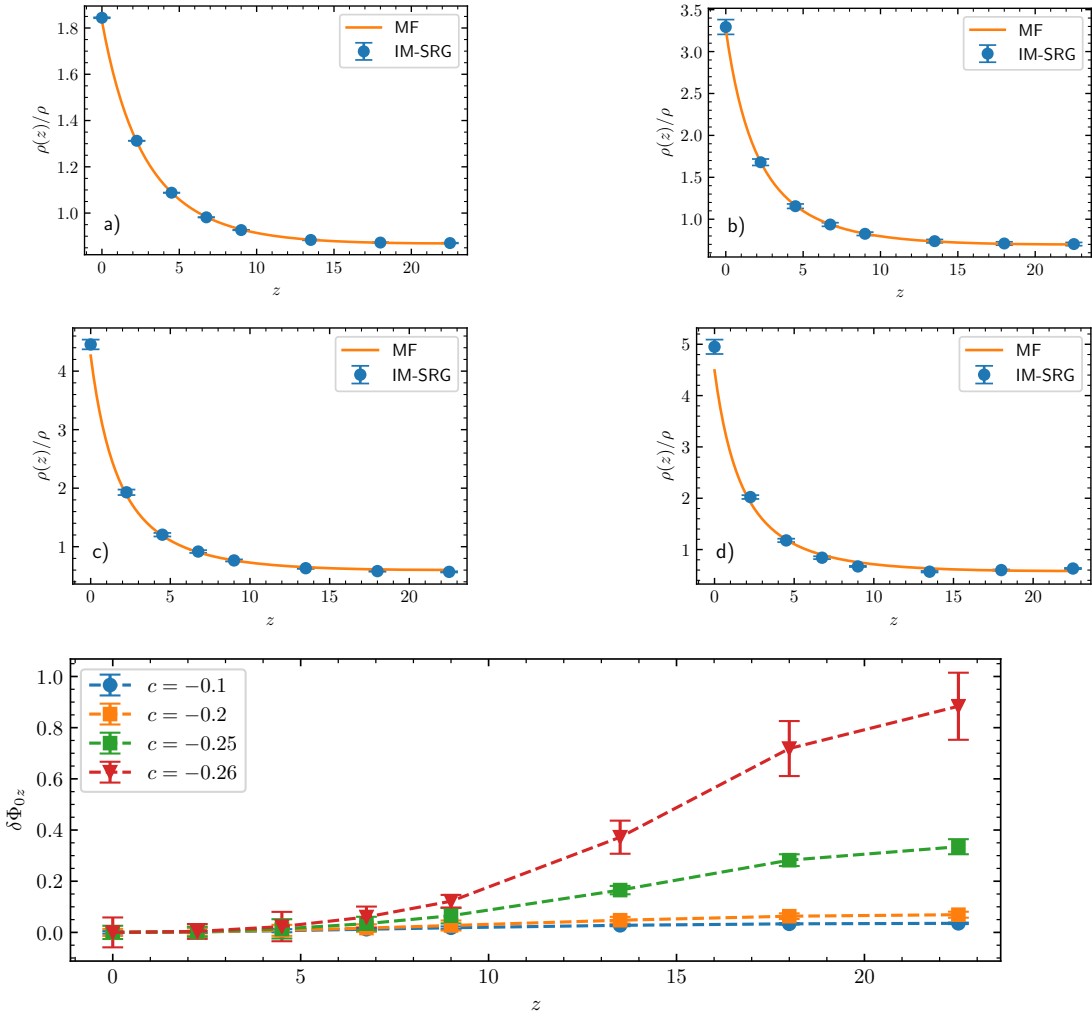

Figure 15: Density of the Bose gas for a heavy attractive impurity ($m \to \infty$, $c < 0$). Blue dots are calculated with the IM-SRG method. The mean-field density is shown as a solid orange curve. The data are for $N = 45$, $\gamma = 0.02$ and a): $c = -0.1$, b): $c = -0.2$, c): $c = -0.25$, d): $c = -0.26$. Panel e) shows the corresponding phase fluctuations. Dots with error bars are calculated using the IM-SRG method. The dashed curves are added to guide the eye.

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
