# Peer review of "Impurities in a one-dimensional Bose gas: the flow equation approach"

_SciPost Physics, doi:SciPost Phys. 11, 008 (2021)_

## Round 2 · Referee Report · Anonymous (Referee 1) · 2021-3-1

Strengths

1 - the problem is interesting in relation to current experiments with polarons in ultracold atoms
2 - the proposed method allows calculation of the spatial properties
3 - a good agreement is found to the known results which serves as a verification of the method

Weaknesses

1 - I find that the Authors did not pay enough attention to make the Manuscript clearly written. The quality of the text and figures can be improved.

Report

Authors study the problem of one or two impurities in a one-dimensional Bose gas. It is proposed to use the renormalization group method in real space for the calculation of the density profile and induced interactions. The validity of the method is demonstrated in comparison with known results. The accuracy of perturbation theories is analyzed, both for repulsive and attractive impurities. The obtained results are relevant to the present experiments with ultracold dilute gases.

The considered problem is important in the context of recent experiments with polarons. The Authors contribute to the progress in the field and the proposed method can be applied to different systems. I find that the obtained results deserve an eventual publication in Sci. Post. journal. But before that a number of issues have to be settled down.

Requested changes

Major comments 1) The manuscript contains a large number of figures and many of them (Figs 1, 2, 6, etc) are not readable in black and white version. I suggest the Authors modify the figures and check if they are easy to read in a printed version. 2) I think that notation f_1 and f_2 is not as obvious as it would be with f_{2b} and f_{GP} denoting two-body and Gross-Pitaevskii solutions used as input for generating the correlations. 3) The ratios “g/ρ” and “c/ρ” which are used to quantify the interaction strength are used as dimensionless parameters, while this is not the case. Here “g” has units of the coupling constant and “ρ” of the density. 4) “One can show that the pair correlation function of the Lieb-Liniger model, g2, is identical to the density of the bosons…” I disagree, for example, for c= ∞ and γ=∞, the pair correlation function g2 = 1 - sin(πρz) ²/ (πρz) ² has oscillations which are always below the asymptotic density while the density of bosons has oscillations which exceed the bulk density. 5) “In our studies, we have noticed that the reference state f2 allows us to investigate a larger range of parameters…” Please specify what exactly is meant by “allowing to investigate” 6) Fig 4c, there is a non-monotonic dependence that looks rather spurious. Is it possible to reduce the errorbars ? 7) Discussion below Fig. 5. “For attractive interactions, the agreement is only quantitative for large values of |c|”, this statement contradicts what is shown in Fig. 5. The agreement is not perfect, but reasonable for repulsive interactions. Exactly the same can be said about the attractive case. In order to keep the claim, I guess, one has to go to a stronger attraction 8) Please comment on the values of C3 in the fit to the energy. Is that a linear or quadratic behavior? As an overall observation, the text is ambiguous in a number of places, is often repetitive “results agree with … results”

Minor comments: 9) Abstract: “observables other than the energy. As an example, we calculate the energies”. Either remove “the energies” or rephrase. 10) Abstract “for two static impurities, we calculate .. correlations”. “Correlation functions” would sound more appropriate, in any case, expand which correlation functions are actually calculated. 11) Abstract does not contain references to renormalization group methods, this has to be stated explicitly 12) Introduction “when solving a Bose-polaron problem on is interested only in local properties …”. And if I am still interested in non-local properties, then what? 13) “The IM-SRG was previously applied to study Bose gases with large condensate fraction”. Is that a 3D system? As the article talks about 1D gas, this should be specified. 14) Approximations are implemented -> assumed 15) I am not sure that it is important to say that Python was used to perform the simulations, the text was written in Latex, etc. For me, mentioning that Runge-Kutta method was used is sufficient. 16) “function which, we discuss” I am not sure you can discuss a function, please rephrase. 17) “the Lieb-Liniger parameter γ and the healing length ξ = 1/ (γ^{1/2} ρ)”. The provided expression applies only for small values of γ, this has to be specified. 18) “The function g2 is known for mesoscopic systems” rephrase 19) “We estimate that ξρ = 7.1”, not clear what “estimate” means here, xi is known exactly 20) “The condensate fraction is a global quantity.” Please rephrase, I am not sure I understand what exactly is meant and why this sentence is here. 21) “we do not present phase fluctuations”, rephrase 22) Figure 5. I suggest putting “c<0” and “c>0” labels inside of the figures 23) Figure 7. I suggest putting “c_2 = …” labels inside of the figures 24) Figure 7, it seems that the errorbars are overestimated. 25) Sections 4.2 and 4.4 “Approach to the thermodynamic limit”. The title is ambiguous. What is “approach” here? Does it refer to a method that describes the thermodynamic or rather to approaching the thermodynamic limit? 26) “We compare the potential … with calculations”, rephrase 27) Figure 9, use the same vertical axis limits in (a) and (c), and (b) and (d). Add “c_2=…” to the legend 28) “each boson is attached to the impurity”. Please explain in more detail. 29)“to understand the induced interactions”, rephrase 30) “The full mean-field solution seems to be cumbersome”. Please specify what is actually meant by that.

I suggest to carefully read the Manuscript, checking what is actually written and if this is what was really intended. The ill-formed phrases and ambiguities do not help to an easier comprehension. As well, it might worth it to give it to read to someone else.

---

## Round 2 · Referee Report · Anonymous (Referee 2) · 2021-3-28

Strengths

see report

Weaknesses

see report

Report

The present paper discusses the applicability of a renormalization group approach to the polaron and bi-polaron problem in weakly interacting one-dimensional Bose gases. The method developed in nuclear physics has recently been successfully extended by some of the authors to weakly interacting Bose gases and to calculate the self energy of the Bose-polaron, a single impurity embedded in the gas. In the present paper the authors develop the approach further and discuss extensions to observables other than the energy as well as to more involved problems, in particular the bi-polaron problem, i.e. the interaction of two impurities mediated by the Bose gas. The paper has two major messages. The first is the discussion of a specific problem, the polaron and bi-polaron in a 1D Bose gas. The second is of methodological nature and concerns the applicability and limitation of the flow-equation approach. While with respect to the first aspect the paper is well written and contains interesting results that deserve publication, I do have some problems with the second. Here considerable revision is needed, both in terms of presentation and in terms of substantiating claims.

Concerning the physics part of the paper

The bi-polaron problem in one dimensional Bose gases has recently regained considerable attention and the authors calculate the interaction energy and interaction potential, as well as the density and phase fluctuations of the quasi-condensate around two impurities. The impurity positions are fixed, i.e. the Born-Oppenheimer limit is considered. The authors find sizable deviations from previous perturbative results when the impurity-boson interaction becomes strong. They also discuss the system-size dependence of observables and point out their relevance for the analysis of cold gas experiments. The results obtained with the flow equation approach agree very well with those obtained from a mean field approximation.

Some comments:

(1) The flow equation results are used to benchmark the mean-field results. There is however no proof or at least some convincing arguments why the flow equation approach should be superior to the mean-field approximation. The authors compare the flow-equation results along with mean-field results with exact data obtained from Bethe ansatz in the integrable case. It would be much more convincing if a parameter case could be found, where there is a sizable difference between flow equations and mean field. Perhaps the author can comment on this.

(2) The authors have calculated the impurity-impurity interaction potential. Ref.[23] predicts an exponential behavior at short distances and a power-law at large distances. The large-distance scaling is due to Casimir-like forces induced by phonon exchange and cannot be captured in the mean field approach. It would be interesting to see if the flow equation approach follows the mean field results or is actually able to reproduce the Casimir forces. Furthermore other recent work in Ref.[76] seems to suggest a linear interaction potential at short distances. Fig. 9 seems to show something different. Can the authors comment on this?

(3) What is the reason that the case of attractive impurity-boson interaction is not included. While one could expect that the mean-field predictions are less accurate here, in contrast to the statement just before eq.(29), the mean-field solutions can be obtained similarly to the repulsive case and it would be interesting to see if the flow equations predict different results than the mean-field approach.

(4) In the extrapolation of the bi-polaron energies to the thermodynamic limit obtained either from flow-equations or mean-field in Fig. 9c and d it appears that the flow equations predict an oscillatory correction on top of the mean field result. This should be discussed.

Concerning the methodological aspect of the paper

The authors use the example of a bose polaron and bi-polaron to illustrate the power of their renormalization approach to weakly interacting Bose gases and use the results as benchmarks for a mean-field theory. While it is certainly very desirable to have methods at hand that allow to tackle these type of problems beyond the mean field level, the authors have not convinced me that this is actually the case for their flow-equation approach. As stated in point (2), the paper would substantially gain if at least some arguments could be provided that the flow equations capture effects beyond the mean-field level. This could be achieved e.g. by comparing to exact solutions, e.g. from Bethe ansatz or from Luttinger liquid theory (see e.g. Ref.[29]), or to QMC simulations, e.g. from Ref.[14] for larger values of the Lieb-Liniger parameter of the Bose gas.

Some comments on the presentation:

The introduction of the renormalization scheme in Sec. 2 is not self contained and thus incomprehensible for a reader unfamiliar with the renormalization scheme. Here the presentation should be improved.

(5) It would be beneficial to explain in a few words what the goal of the unitary transformations governed by the flow equation (1) is. What is meant if the authors say: "to choose the operatore eta(s) .... to steer the flow in the desired direction."? What desired direction?

(6) Right after eq.(2) the reference state is mentioned without any introduction. The latter is only given in the following subsection.

(7) Normal ordering with respect to the reference state is mentioned but only defined in the Appendix.

(8) The flow equation results show error bars from "relative truncation error". Reference to Appendix A.4 should be made here to explain how the truncation error was obtained.

In summary the manuscript contains interesting and novel results warranting in principle publication but needs revision.

Requested changes

see report

---

## Round 3 · Referee Report · Anonymous · 2021-6-11

Report

I find that most of my comments were adequately taken into account and I recommend a publication once my last comment is answered.

Requested changes

As a final remark, I strongly suggest using the full units in the figures and equations. Although the choice hbar=1 and m=1 is often used, it does not allow to use the criteria of proper units for checking the correctness of a certain expression.

---

## Round 3 · Referee Report · Anonymous · 2021-6-14

Strengths

1) benchmark of mean-field results for Bose polaron and bi-polaron in 1D gas
2) extends previous work of authors to different observables such as density profile of Bose gas and interaction potential

Weaknesses

1) bipolaron problem discussed only for asymmetric case of one impurity interacting infinitely strongly and the second with variable strength
2) benefit of the flow equation approach becomes only apparent after consulting earlier work of the authors

Report

The authors have essentially addressed all the points of my previous report and I recommend publication after they consider the following optional comments.

Requested changes

1) Oscillations in the impurity-impurity potential in Fig.9b: The authors state at the end of page 19 that (i) they expect a monotonous increase and (ii) that it is difficult for them to estimate the error bars for the curve in the thdyn. limit. All curves in Fig. 9a and 9b are shown with error bars, except the curves in the thdn. limit. On first glance the crosses suggest however that there are error bars plotted as well. To avoid confusion I suggest to explicitly mention in the figure caption that no error bars are given here since their estimation is difficult.

2) The arguments given by the authors in the text and in Appendix B for why attractive impurity-boson interactions are not treated in detail and referred to a future publication are not convincing. The comparision of mean-field contact parameter in Fig.5a for c <0 does not appear much worse than in Fig.5b for c>0. The same applies to Fig.15 a-d. The most convincing argument for me is the plot of the phase fluctuations in Fig. 15e. I suggest to refer to this plot when arguing that the attractive case requires more careful analysis.

3) The definition of normal ordering with respect to the reference state, now explained a bit more in detail in Appendix A.1, is simple enough and yet of sufficient importance for a non-specialist reader to understand the idea of the approach, that it should be put into th main text of the paper.

4) In the introduction the authors say at the beginning of page 3 that it is of particular interest to compare their flow equation approach with Wilson-type RG techniques and that the IM-SRG complements that technique. I did not quite understand this comment. It seems that the Wilson-type RG approach gives results which deviate substantially already from the mean-field result of the authors, see paragraph before Sec. 3.3.

5) Finally it would be helpful to add to the legend (not the figure caption) in Fig.3 that the reference state in the left figure is f_GP and in the right figure it is f_1b.

---

## Round 3 · Author Response

We thank the Referees for taking the time to review our paper. We are happy to see the overall positive evaluation of our paper. The comments in the reports helped us to significantly improve our manuscript. We hope that the revised version is ready for publication in SciPost Physics.

Below, we provide a point-by-point reply to the comments in the reports. For convenience, we quote the comments of the Referees. The main changes in the manuscript are shown in blue. Minor changes are not highlighted in the manuscript.

Reply to Anonymous Report 1:

Major comments:

  1. Referee: "The manuscript contains a large number of figures and many of them (Figs 1, 2, 6, etc) are not readable in black and white version. I suggest the Authors modify the figures and check if they are easy to read in a printed version."

Our reply: We changed all figures which were not readable in black and white. We modified the captions accordingly.

  1. Referee: "I think that notation f_1 and f_2 is not as obvious as it would be with f_{2b} and f_{GP} denoting two-body and Gross-Pitaevskii solutions used as input for generating the correlations."

Our reply: We thank the Referee for this useful suggestion. We have modified the text and figures accordingly. Note that we chose to use f_{1b} instead of the suggested f_{2b} to emphasize that the reference state is the solution of a one-boson Hamiltonian.

  1. Referee: "The ratios “g/ρ” and “c/ρ” which are used to quantify the interaction strength are used as dimensionless parameters, while this is not the case. Here “g” has units of the coupling constant and “ρ” of the density."

Our reply: Please note that we are using the system of units in which hbar=M=1. This implies that g/rho and c/rho are dimensionless.

We added footnote 4 to clarify this point:

System Message: WARNING/2 (<string>, line 27)

Definition list ends without a blank line; unexpected unindent.

"In general, the dimensionless Lieb-Liniger parameter is defined as gamma=Mg/(hbar^2rho), which leads to gamma=g/rho in our units (hbar=M=1)."

  1. Referee: "“One can show that the pair correlation function of the Lieb-Liniger model, g2, is identical to the density of the bosons…” I disagree, for example, for c= ∞ and γ=∞, the pair correlation function g2 = 1 - sin(πρz) ²/ (πρz) ² has oscillations which are always below the asymptotic density while the density of bosons has oscillations which exceed the bulk density."

Our reply: We thank the Referee for alerting us of this potentially confusing statement. Please note that the density of bosons is measured in the frame co-moving with the impurity. We have modified the text to stress this point and to clarify the relation between pair correlation function and density in the co-moving frame, see the discussion around Eqs. (17-19) of the revised manuscript. We have also identified a typo in our formula. The revised text presents the correct expression.

  1. Referee: "“In our studies, we have noticed that the reference state f2 allows us to investigate a larger range of parameters…” Please specify what exactly is meant by “allowing to investigate”"

Our reply: We thank the Referee for identifying this imprecise statement. To clarify it, we have re-written the discussion, which now reads as:

“In our studies, we noticed that the reference state f_{GP} is generally a better choice than f_{1b}. In comparison to IM-SRG(f_{1b}), the scheme IM-SRG(f_{GP}) allows us to obtain converged results for a larger range of parameters. In particular, IM-SRG(f_{GP}) is more reliable for large systems, and large boson-boson interactions.”

  1. Referee: "Fig 4c, there is a non-monotonic dependence that looks rather spurious. Is it possible to reduce the errorbars?"

Our reply: Unfortunately, it is not possible to reduce the error-bars using out truncation scheme. Note that the density of bosons at the position of the impurity is small. Therefore, large error-bars in Fig. 4c is a result of working with small numbers, which cannot be avoided, unless we significantly modify our approach.

In the revised version, we have added a sentence to clarify this statement. See footnote 8: “Note that we expect that the exact curve for c/ρ= 0.5 in Fig. 4 c) is monotonous. Our calculations of thiscurve have large error-bars, which allow for an apparently non-monotonous behavior.”

  1. Referee: "Discussion below Fig. 5. “For attractive interactions, the agreement is only quantitative for large values of |c|”, this statement contradicts what is shown in Fig. 5. The agreement is not perfect, but reasonable for repulsive interactions. Exactly the same can be said about the attractive case. In order to keep the claim, I guess, one has to go to a stronger attraction"

Our reply: We thank the referee for this comment. We have modified the discussion accordingly: “ For c >0, the agreement between Monte-Carlo and the mean-field approximation is reasonable for all available data points. For attractive interactions, the difference between the results is more noticeable, which implies that the MFA leads to less accurate results for c <0, see also Appendix B, where we present some additional data for the case with attractive interactions”

  1. Referee: "Please comment on the values of C3 in the fit to the energy. Is that a linear or quadratic behavior?"

Our reply: We thank the Referee for this suggestion. The revised version now states that the parameter C3 is in between 1 and 2, see footnote 11. To be more precise, we find that for c_2/rho=0.02 C3 is between 1.25 and 1.7 depending on d. For c_2/rho=0.1, C3 is between 1.1 and 2. We also checked that these windows are consistent with the value obtained by fitting to MF results. We noticed that large values of d usually imply smaller values of C3.

Minor Comments: We thank the Referee for providing us with minor comments [The revised version addresses all of them.]. They helped us to significantly improve our manuscript.

Reply to Anonymous Report 2

Referee Report 2:

  1. Referee: "The flow equation results are used to benchmark the mean-field results. There is however no proof or at least some convincing arguments why the flow equation approach should be superior to the mean-field approximation. The authors compare the flow-equation results along with mean-field results with exact data obtained from Bethe ansatz in the integrable case. It would be much more convincing if a parameter case could be found, where there is a sizable difference between flow equations and mean field. Perhaps the author can comment on this."

Our reply: We thank the Referee for alerting us of this weakness in interpreting our results. Indeed, the flow equations are more accurate than the mean-field approximation. We had demonstrated this in our previous works, for example, by calculating the energy of the Lieb-Liniger model, see [1] in the revised manuscript. In that paper, for gamma=1 and Nsim 10, the flow equations yield the exact energy with 10% accuracy, the mean-field energy for the same parameters is about 50% larger.

In the revised version, we address this issue in the Introduction: “IM-SRG was recently extended to cold Bose gases [1, 2]. It was tested by calculating the ground-state energies of the Lieb-Liniger model and a one-dimensional (1D) Bose gas with an impurity atom (‘Bose polaron’) [1, 2]. Those works demonstrate that flow equations allow one to go beyond mean-field approximation without relying on many-body perturbation theory. In the present work, we use IM-SRG to calculate the density and phase fluctuations of the Bose gas."

In addition, following the recommendation of the Referee, we illustrate a parameter regime for which there is a sizable difference between flow equations and mean field, see new Appendix B in the revised manuscript. In particular, we demonstrate that for an attractive impurity the flow equations predict significant phase fluctuations, which are beyond the mean-field approximation.

  1. Referee: "The authors have calculated the impurity-impurity interaction potential. Ref. [23] predicts an exponential behavior at short distances and a power-law at large distances. The large-distance scaling is due to Casimir-like forces induced by phonon exchange and cannot be captured in the mean field approach. It would be interesting to see if the flow equation approach follows the mean field results or is actually able to reproduce the Casimir forces. Furthermore other recent work in Ref.[76] seems to suggest a linear interaction potential at short distances. Fig. 9 seems to show something different. Can the authors comment on this?"

Our reply: We absolutely agree, a numerical validation of the long-range Casimir-like force would be an extremely interesting result. Unfortunately, that force is very weak. The Casimir-like force is important only at distances of the order of 5-10 xi (xi for the healing length), which are larger than the typical sizes we consider. We have added a corresponding remark to the manuscript, see also very recent works on the topic in Refs. [76,77], where a similar conclusion is reached.

To the best of our knowledge, currently, there are no numerical techniques capable of calculating the Casimir-like force. In particular, it is also out-of-reach of the state-of-the-art Quantum Monte Carlo results [76].

Please note that 2101.11997 suggests linear interaction only for etatoinfty, i.e., not for the case presented in Fig. 9. We also expect a linear behavior when c_2 becomes very large, since our equations are identical to those of 2101.11997. The case of large c_2 is however not discussed in our work.

  1. Referee: "What is the reason that the case of attractive impurity-boson interaction is not included. While one could expect that the mean-field predictions are less accurate here, in contrast to the statement just before eq.(29), the mean-field solutions can be obtained similarly to the repulsive case and it would be interesting to see if the flow equations predict different results than the mean-field approach."

Our reply: We thank the Referee for this comment. Our plan is to consider the system with attractive boson-impurity interactions in another publication. For finite systems, the attractive case is fundamentally different from the repulsive case. In particular, there is a transition from a many-body bound state to a state in which the bosons may occupy scattering states. This transition requires a separate discussion, which is beyond the scope of the present paper. Following the recommendation of the Referee, we have added Appendix B to the revised manuscript. The Appendix shows some data for a Bose gas with a single attractive impurity. In particular, it illustrates significant phase fluctuations that happen close to the aforementioned transition.

  1. Referee: "In the extrapolation of the bi-polaron energies to the thermodynamic limit obtained either from flow-equations or mean-field in Fig. 9c and d it appears that the flow equations predict an oscillatory correction on top of the mean field result. This should be discussed."

Our reply: We thank the Referee for this remark. We believe that the oscillating behavior is due to the numerical accuracy of our results. In particular, the amplitude of oscillations is within the error-bars for N=70. Unfortunately, it is not easy for us to increase the accuracy for these large systems.

We have added a clarifying discussion to the revised manuscript (see page 19): “The truncation error in the IM-SRG method grows rapidly with the number of particles.This rapid growth rules out a reliable extrapolation of the errorbars to the thermodynamic limit. Therefore, we give no estimate for the accuracy of C1, which leads to an apparently oscillating character of the potential in the thermodynamic limit. We expect that the exact potential is a monotonically increasing function of the distance between the impurities,d.”

  1. Referee: "It would be beneficial to explain in a few words what the goal of the unitary transformations governed by the flow equation (1) is. What is meant if the authors say: "to choose the operatore eta(s) .... to steer the flow in the desired direction."? What desired direction?"

Our reply: Following the recommendation of the Referee, we have added a corresponding discussion to the revised manuscript. See the discussion around Eqs. (1) and (2):

“[…]which transforms the Hamiltonian matrix into a block-diagonal form, i.e., it decouples the“ground-state” matrix element from all excitations (see Fig. 1).”

“ In our work, eta(s) is chosen from the matrix elements that describe the couplings between the ‘condensate’ and its excitations such that these couplingsbecome weaker as the flow progresses, see Fig. 1. A detailed construction of eta(s) is presented in Appendix A.”

  1. Referee: "Right after eq.(2) the reference state is mentioned without any introduction. The latter is only given in the following subsection."

Our reply: We thank the Referee for this comment. We have modified the text to make the discussion more clear. In particular, we no longer mention the reference state in Chap. 2.1 “Flow equations”. Instead, we moved the discussion of our truncation scheme (in which we mentioned the reference state) to Chap. 2.2 “Reference state”.

  1. Referee: "Normal ordering with respect to the reference state is mentioned but only defined in the Appendix."

Our reply: Following the recommendation of the Referee, in the revised version, we make a more clear reference to the definition of normal ordering in the Appendix.

  1. Referee: "The flow equation results show error bars from "relative truncation error". Reference to Appendix A.4 should be made here to explain how the truncation error was obtained."

Our reply: We thank the Referee for this omission in the original version of our manuscript. We have added a discussion which clarifies the calculation of errorbars, and makes a more explicit reference to the Appendix.

Chapter 2.2 now starts as: “In general, it is impossible to solve Eq. (1) for a many-particle system without approximations. The complexity is due to the commutator [η,H]: It leads to many-body terms, which are not present in the initial Hamiltonian H(s= 0). To solve Eq. (1), the many-body terms must be truncated at some order. To define a truncation hierarchy, we write the Hamiltonian in second quantization using normal ordering with respect to a reference state (for a definition of normal ordering see Appendix A.1). The reference state, Ψref, should approximate an eigenstate (here the ground state) of the Hamiltonian well, otherwise the IM-SRG transformation cannot map Ψref onto the exact state. Upon normal ordering, we truncate three-body excitations and beyond, see Fig. 1. To estimate the introduced truncation error, we use the three-body elements and second order perturbation theory for matrices, see Fig. 1 and Appendix A.2.”

Docutils System Messages

System Message: ERROR/3 (<string>, line 50); backlink

Undefined substitution referenced: "c".

---

## Round 4 · Author Response

Dear Editor,

thank you for communicating to us the editorial decision on our manuscript on June 14th.

We have revised the manuscript in accord with the second reports of the Referees.
We hope that the revised version of our manuscript is ready for publication in SciPost Physics.

Best wishes,

Artem Volosniev, on behalf of the authors

---

## Round 4 · List of Changes

We thank the Referees for reviewing our work. The revised version of the manuscript addresses all comments of the Referees.
For convenience of the reader, we highlight major changes in the manuscript. In addition, we provide a list of changes below.
* * *
List of Changes in Response to Anonymous Report 3

1. The Referee "Oscillations in the impurity-impurity potential in Fig.9b: The authors state at the end of page 19 that (i) they expect a monotonous increase and (ii) that it is difficult for them to estimate the error bars for the curve in the thdyn. limit. All curves in Fig. 9a and 9b are shown with error bars, except the curves in the thdn. limit. On first glance the crosses suggest however that there are error bars plotted as well. To avoid confusion I suggest to explicitly mention in the figure caption that no error bars are given here since their estimation is difficult."

Our reply: We thank the Referee for this remark. The revised captions of Figs. 9 and 12 state explicitly that we do not present error bars for the curves in the thermodynamic limit. To avoid any confusion, we have also used another symbol to present our data in the thermodynamic limit.

2. The Referee: "The arguments given by the authors in the text and in Appendix B for why attractive impurity-boson interactions are not treated in detail and referred to a future publication are not convincing. The comparision of mean-field contact parameter in Fig.5a for c <0 does not appear much worse than in Fig.5b for c>0. The same applies to Fig.15 a-d. The most convincing argument for me is the plot of the phase fluctuations in Fig. 15e. I suggest to refer to this plot when arguing that the attractive case requires more careful analysis."

Our reply: To address this comment of the Referee, we have modified the discussion in Sec. 3.4. In particular we added a sentence: "Note in particular Fig. 15, which indicates large phase fluctuations for moderate impurity-boson interactions, in contrast to the repulsive case."

3. The Referee: "The definition of normal ordering with respect to the reference state, now explained a bit more in detail in Appendix A.1, is simple enough and yet of sufficient importance for a non-specialist reader to understand the idea of the approach, that it should be put into th main text of the paper."

Our reply: To address this comment of the Referee, we moved our discussion on normal ordering to the main text, see Sec. 2.2.

4. The Referee: "In the introduction the authors say at the beginning of page 3 that it is of particular interest to compare their flow equation approach with Wilson-type RG techniques and that the IM-SRG complements that technique. I did not quite understand this comment. It seems that the Wilson-type RG approach gives results which deviate substantially already from the mean-field result of the authors, see paragraph before Sec. 3.3."

Our reply: We thank the Referee for pointing out this potentially confusing statement. The Wilson-type RG technique is formulated in momentum space. Our work is formulated in real space. The difference between our result and the Wilson-type RG indicates that the Bose polaron problem can be studied more accurately in the real-space formulation. To avoid any confusion, we moved this discussion from the introduction to footnote 6.

5. The Referee: "Finally it would be helpful to add to the legend (not the figure caption) in Fig.3 that the reference state in the left figure is f_GP and in the right figure it is f_1b."

Our reply: We have modified the figures accordingly.
* * *
List of changes in Response to Anonymous Report 1

The Referee: "As a final remark, I strongly suggest using the full units in the figures and equations. Although the choice hbar=1 and m=1 is often used, it does not allow to use the criteria of proper units for checking the correctness of a certain expression."

Our reply: We thank the Referee for this suggestion. To address this comment, we have introduced proper dimensionless quantities at the beginning of Sec. 3, and have modified all equations and figures accordingly.

---

## Editorial Decision

published